DISCOVERY REPORT

# The X-linked splicing regulator MBNL3 has been co-opted to restrict placental growth in eutherians

**Thomas Spruce**[1]*, **Mireya Plass**[1,2,3], **André Gohr**[1], **Debashish Ray**[4], **María Martínez de Lagrán**[1], **Gregor Rot**[5], **Ana Nóvoa**[6], **Demian Burguera**[1], **Jon Permanyer**[1], **Marta Miret**[1], **Hong Zheng**[4], **Maurice S. Swanson**[7], **Quaid Morris**[4], **Moises Mallo**[6], **Mara Dierssen**[1,8,9], **Timothy R. Hughes**[4], **Barbara Pernaute**[1]*, **Manuel Irimia**[1,8,10]*

1 Centre for Genomic Regulation, Barcelona Institute of Science and Technology, Barcelona, Spain, 2 Gene Regulation of Cell Identity, Regenerative Medicine Program, Bellvitge Institute for Biomedical Research (IDIBELL), Barcelona, Spain, 3 Program for Advancing Clinical Translation of Regenerative Medicine of Catalonia, P-CMR[C], L'Hospitalet del Llobregat, Barcelona, Spain, 4 Donnelly Centre for Cellular and Biomolecular Research, University of Toronto, Toronto, Ontario, Canada, 5 Institute of Molecular Life Sciences of the University of Zurich and Swiss Institute of Bioinformatics, Zurich, Switzerland, 6 Instituto Gulbenkian de Ciência, Oeiras, Portugal, 7 Department of Molecular Genetics and Microbiology, Center for NeuroGenetics and the Genetics Institute, College of Medicine, University of Florida, Gainesville, Florida, United States of America, 8 Universitat Pompeu Fabra, Barcelona, Spain, 9 Centro de Investigación Biomédica en Red de Enfermedades Raras, Barcelona, Spain, 10 ICREA, Barcelona, Spain

* thomas.spruce@crg.eu (TS); barbara.pernaute@crg.eu (BP); mirimia@gmail.com (MI)

The Editors encourage authors to publish research updates to this article type. Please follow the link in the citation below to view any related articles.

## Abstract

Understanding the regulatory interactions that control gene expression during the development of novel tissues is a key goal of evolutionary developmental biology. Here, we show that *Mbnl3* has undergone a striking process of evolutionary specialization in eutherian mammals resulting in the emergence of a novel placental function for the gene. *Mbnl3* belongs to a family of RNA-binding proteins whose members regulate multiple aspects of RNA metabolism. We find that, in eutherians, while both *Mbnl3* and its paralog *Mbnl2* are strongly expressed in placenta, *Mbnl3* expression has been lost from nonplacental tissues in association with the evolution of a novel promoter. Moreover, *Mbnl3* has undergone accelerated protein sequence evolution leading to changes in its RNA-binding specificities and cellular localization. While *Mbnl2* and *Mbnl3* share partially redundant roles in regulating alternative splicing, polyadenylation site usage and, in turn, placenta maturation, *Mbnl3* has also acquired novel biological functions. Specifically, *Mbnl3* knockout (M3KO) alone results in increased placental growth associated with higher *Myc* expression. Furthermore, *Mbnl3* loss increases fetal resource allocation during limiting conditions, suggesting that location of *Mbnl3* on the X chromosome has led to its role in limiting placental growth, favoring the maternal side of the parental genetic conflict.

## Introduction

Placentas have evolved independently many times in vertebrates [1] and provide a direct link between mother and fetus through which nutrient exchange can occur. In eutherians, the

**Data Availability Statement:** All relevant data are within the paper and its Supporting Information files. In particular, all transcriptomic data generated for the study and the associated SRA IDs are listed in S1 Table.

**Funding:** The research has been funded by the Spanish Ministerio de Ciencia (BFU2017-89201-P and PID2020-115040GB-I00 to M.I.), the 'Centro de Excelencia Severo Ochoa 2013-2017'(SEV-2012-0208 to the Centre for Genomic Regulation), and the People Programme (Marie Curie Actions) of the European Union's Seventh Framework Programme (FP7/2007-2013) under REA grant agreement n° 608959 to T.S. The funders had no role in study design, data collection and analysis, decision to publish, or preparation of the manuscript.

**Competing interests:** The authors have declared that no competing interests exist.

**Abbreviations:** CEA, carcinoembryonic antigen; CEEA-PRBB, Comitè Ètic d'Experimentació Animal-Parc de Recerca Biomèdica de Barcelona; DKO, double knockout; ECDF, empirical cumulative distribution function; epiDKO, epiblast-specific Mbnl2 KO and universal Mbnl3 KO; ESC, embryonic stem cell; FDR, false discovery rate; GO, Gene Ontology; ICM, inner cell mass; KO, knockout; M3KO, Mbnl3 knockout; MBNL, Muscleblind-like; PSI, percentage spliced in; RNA-seq, RNA sequencing; SRA, Short Read Archive; TE, trophectoderm; TF, transcription factor; TPM, transcript per million; TS, trophoblast stem; TSS, transcription start site; XEN, extraembryonic endoderm; WT, wild-type.

placenta is mainly formed from the trophoblast, a mammalian-specific innovation derived from the embryo in the first lineage decision of development. Placental evolution results in the loss of full maternal control of fetal resource allocation via yolk formation. In species where females breed with multiple males over their lifetimes, this may result in the placenta becoming a site of parental conflict over resource allocation [2]. Such conflicts are proposed to occur because, in these species, siblings are more likely to be related via their maternal than their paternal genome. This means that the inclusive fitness of paternally inherited alleles is optimal when there is a high level of resource allocation to the individual offspring containing them. On the other hand, the inclusive fitness of maternally inherited alleles is optimized when there is a greater degree of maternal resource sharing between siblings and thus lower levels of maternal investment in individual offspring [3]. Genetic conflicts over resource allocation have likely contributed to the accumulation of genes with placentally biased expression on the X chromosome [4] as sex chromosomes are expected to accumulate sexually antagonistic genes due to their asymmetric inheritance pattern and distribution [5–7]. This effect is likely enhanced in species such as mouse, where trophoblastic gene expression from the X chromosome is strongly paternally imprinted and occurs almost exclusively from maternal alleles [8], which is likely to drive the accumulation of maternally favorable variants reducing resource allocation [9–11]. Indeed, in mouse, placental hyperplasia is seen when a number of placentally expressed X-linked genes are knocked out [12–14].

To date, understanding the regulation of protein production in the eutherian placenta has mainly been focused at the transcriptional level [15,16]. In contrast, with the exception of miRNA-mediated modulation of gene expression [17], little is known about the role of posttranscriptional regulation of processes such as alternative splicing, alternative polyadenylation, and mRNA translation. In particular, alternative splicing, in which an exon or intron is either excluded from or included in a transcript, has the potential to greatly expand both transcriptomic and proteomic diversity [18–20]. RNA-binding proteins mediate much of this posttranscriptional regulation, with the same proteins often regulating multiple aspects of RNA metabolism [21,22]. Targeting of RNA by these regulators is usually mediated by RNA-binding domains [23], which are typically discrete and ordered. However, it has recently become apparent that intrinsically disordered regions can also mediate binding [24]. RNA-binding proteins often contain multiple RNA-binding domains, increasing both their binding specificity and affinity [22].

To find posttranscriptional regulators with relevance for placenta development and evolution, we first searched for splicing regulators with enriched expression in trophoblastic versus. nontrophoblastic tissues. We identified 2 Muscleblind-like (MBNL) genes, *Mbnl2* and *Mbnl3*, which, through a combination of comparative genomic approaches and in vitro binding assays, we found to have undergone strikingly different evolutionary paths during the emergence of the eutherian lineage. Using mouse knockout (KO) approaches coupled with high throughput sequencing, we investigated the functional role of these factors in placenta and identified a unique role in restricting placental growth for the X-linked gene *Mbnl3*.

## Results

### *Mbnl3* has the strongest placenta enrichment among splicing factors

In order to identify splicing regulators with a potential placenta-specific function, we began by screening a combination of publicly available and in-house transcriptomic datasets (S1 Table) for splicing factors that are differentially expressed between trophoblastic and embryonic tissues in mouse. The following expression comparisons were made: (i) placenta versus nonplacental tissues; (ii) trophoblast versus inner cell mass (ICM) at the blastocyst stage; and (iii)

trophoblast stem (TS) cells versus embryonic and extraembryonic endoderm (XEN) stem cells, for 197 previously defined splicing regulators (Fig 1A–1C, S2 Table; see Methods [25,26]). These analyses revealed the X-linked gene *Mbnl3* to be substantially more enriched than any of the other regulators examined in both placenta (analysis (i)) and E3.5 trophoblast (analysis (ii)) (Fig 1A and 1B). Six other genes were also identified as being enriched (log2FC > 1) in trophoblastic tissues in at least one of these comparisons: *Mbnl2* (an autosomal *Mbnl3* paralog), *Esrp1*, *Esrp2*, *Rbm20*, *Rbm38*, and *Rbms1*. For the top candidates, trophoblast expression was confirmed by in situ hybridization analysis (Fig 1D, S1A Fig). This showed *Rbm20* to be specifically expressed in the extraembryonic ectoderm adjacent to the epiblast at E6.5 and in the chorionic hinge at late E7.5, regions that have been described as containing TS cells [27–29]. In contrast, expression of *Mbnl3* and *Esrp2* appeared to be restricted to differentiated trophectoderm (TE) tissue and absent from these regions. Finally, *Mbnl2*, *Rbm38*, and *Rbms1* were expressed in both TS cell containing regions and more differentiated TE tissues.

Importantly, of the splicing factors examined, only *Mbnl3* was highly placenta specific, with no or low expression in other tissues (Fig 1E, S1E Fig). Moreover, analysis of RNA sequencing (RNA-seq) data from human and cow showed placental expression of these factors to be highly conserved, with *Mbnl3* also showing the strongest placenta-specific enrichment alongside very low levels of expression in nonplacental tissues (Fig 1F, S1C–S1E Fig, S1, S3, and S4 Tables). In contrast, *Mbnl3* was strongly expressed across a range of adult tissues in noneutherians (S1E Fig, S1 Table), including the marsupial opossum, for which *Mbnl3* showed no placenta-enriched expression (Fig 1F, S1E Fig, S5 Table).

## Placental expression of *Mbnl3* from a novel eutherian-specific promoter

Comparison of the *Mbnl3* gene locus across a number of vertebrate species revealed it harbors a novel eutherian-specific transcription start site (TSS), adjacent to a eutherian-specific hAT family DNA transposon *MER53*, in addition to canonical TSSs, conserved across vertebrates, and more lineage-restricted TSSs (Fig 2A, S2A–S2C Fig). The use of the eutherian-specific TSS, or of rodent or primate specific TSSs found within <5 kbp of it, results in a transcript that codes for an amino-terminally truncated protein (Fig 2B). A similarly truncated MBNL3 protein isoform can also be generated from alternatively spliced transcripts that initiate from the ancestral TSSs but skip exon 2, which encodes the canonical start codon (Fig 2A). Consistent with previous studies, we found that both full-length and truncated MBNL3 protein isoforms were present in mouse placenta by western blot analysis (S2D Fig). To investigate the use of the different TSSs and amino-terminal isoforms, we quantified RNA-seq reads spanning competing exon–exon junctions (a–d in Fig 2A) across a range of tissues from mouse, human, and cow (Fig 2C, S2E and S2F Fig, S6 Table). This analysis revealed the eutherian-specific TSS, alongside the nearby lineage-specific ones, to be dominant in blastocyst stage trophoblast tissue in all 3 species. Furthermore, in placenta, the eutherian-specific TSS was found to account for around half of the transcripts in mouse and almost all of the transcripts in human and cow. Moreover, placenta-specific ChIP-seq data revealed an enrichment for H3K27ac and H3K4me3 active chromatin marks around the eutherian-specific TSS in mouse (S2G Fig). In line with these observations, transgenic reporter assays revealed that this region contains regulatory elements capable of driving trophoblast-specific expression in mouse embryos (Fig 2D).

## MBNL3 has undergone accelerated sequence evolution in eutherians

MBNL proteins contain 2 pairs of zinc fingers (ZF1-2 and ZF3-4), which mediate RNA binding, separated by a linker region, which is required for splicing regulation and that may play a role in RNA binding (Fig 2B [30–34]). The first pair of zinc fingers is missing from the truncated

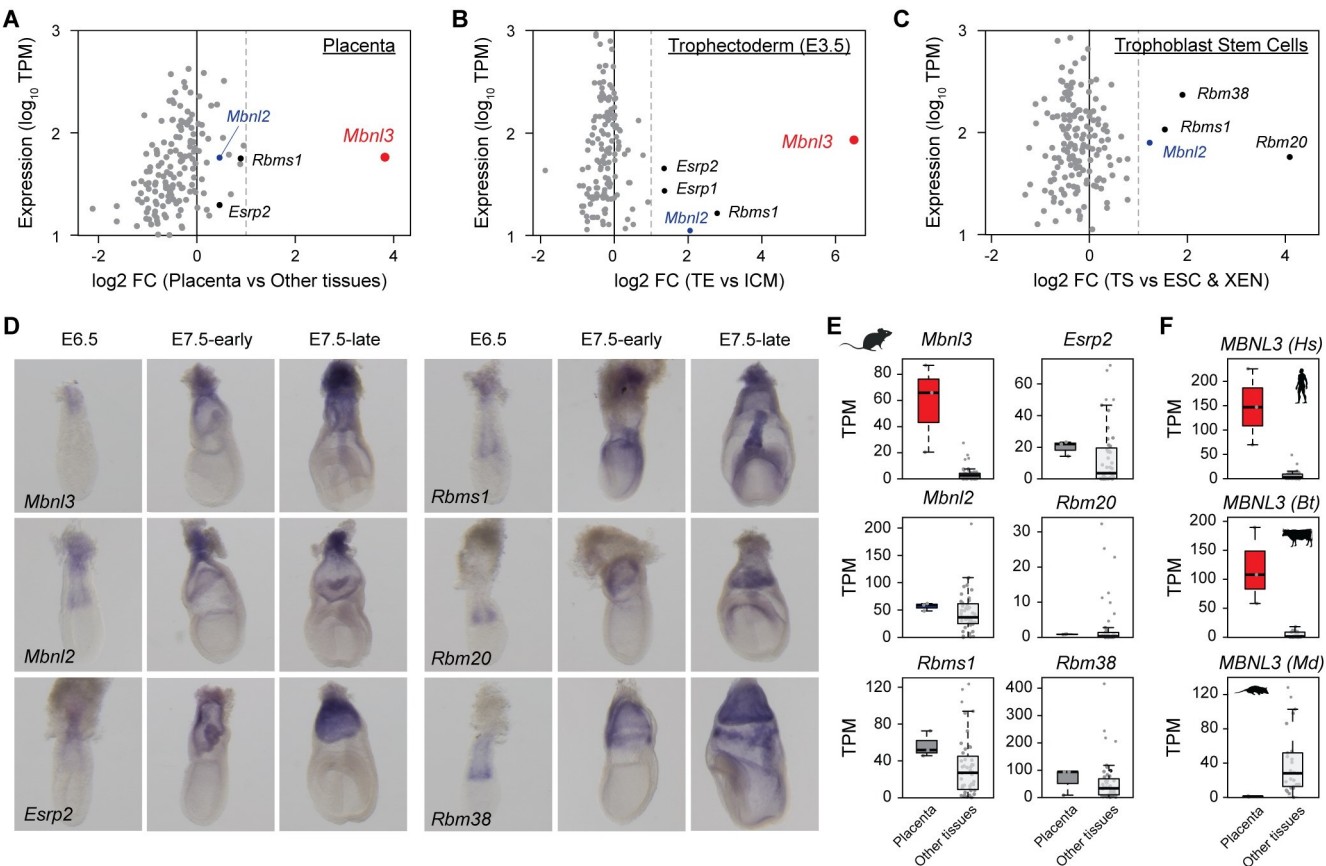

**Fig 1. Analysis of splicing factor expression enrichment in trophoblastic versus nontrophoblastic tissues. (A–C)** Scatter plots showing trophoblastic expression level and enrichment for 197 splicing regulators in 3 separate comparisons of trophoblastic versus nontrophoblastic tissues. **(D)** Whole-mount in situ hybridisation analysis of the expression of splicing regulators found to be enriched in trophoblastic tissues (A–C) in mouse embryos at the indicated stages. **(E, F)** Box plots showing expression levels of the indicated splicing factors in placenta and nonplacental tissues in mouse (E) and in human (Hs), cow (Bt), and opossum (Md) (F). The numerical data underlying this figure can be found in S1 Data. ESC, embryonic stem cell; ICM, inner cell mass; TE, trophectoderm; TPM, transcript per million; TS, trophoblast stem; XEN, extraembryonic endoderm.

eutherian MBNL3 isoform (Fig 2B), and equivalent truncated protein isoforms have not been observed for MBNL1 or MBNL2 [35–37]. Using phylogenetic analysis, we found that *Mbnl3* has undergone accelerated sequence evolution in eutherians compared to noneutherian *Mbnl3* and all *Mbnl2* genes (Fig 2E). In particular, eutherian MBNL3 proteins show specific changes in conserved amino acids in ZF1, the spacers separating ZFs 1 and 2 and ZFs 3 and 4, as well as the linker region (Fig 2B, S3A Fig). In contrast, the protein sequence of MBNL2 is highly conserved across vertebrates. Close examination of *Mbnl3* gene structure identified an additional feature to be associated with eutherians: loss of a 54-bp alternative exon [38], which triggers nuclear localization upon inclusion [33] and that is found in all *Mbnl1* and *Mbnl2* paralogs and in most noneutherian *Mbnl3* genes (S3B Fig). Altogether, these eutherian-specific changes to *Mbnl3*, alongside the loss of its expression from nonplacental tissues and its location in the X chromosome, suggest that it may have undergone a process of evolutionary specialization in this clade.

## Eutherian-specific changes in MBNL3 binding preferences

To investigate how these eutherian-specific novelties may have affected MBNL3 functionality, we first examined the RNA-binding preference of full-length (long) eutherian (mouse) and

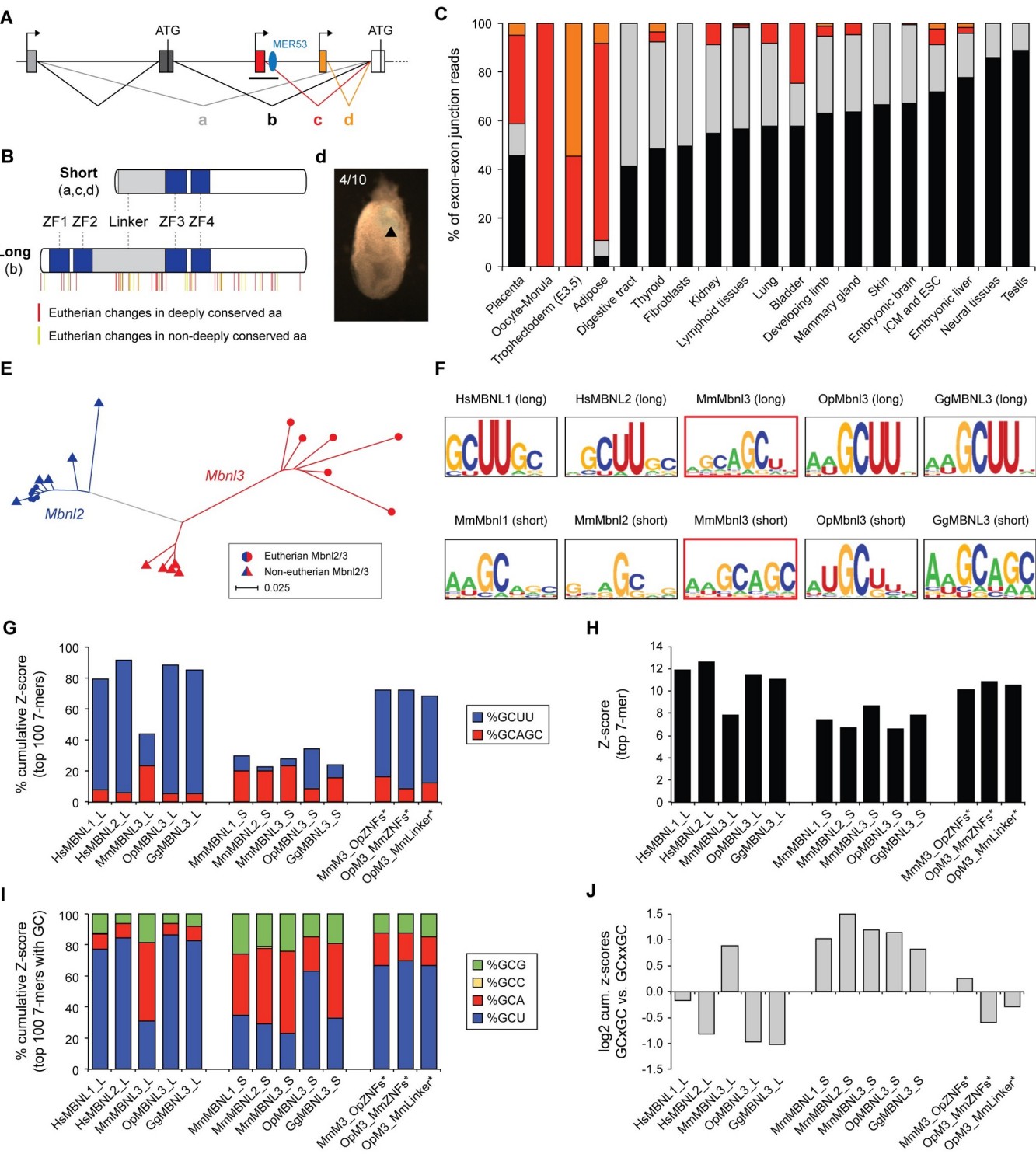

**Fig 2. Molecular evolution of *Mbnl3* in eutherians. (A)** Schematic showing alternative splicing events and transcriptional (arrow) and translational (ATG) start sites leading to long and short MBNL3 isoform production. Splice junctions corresponding to transcripts that code for the long (b) and short (a, c, d) isoforms are indicated. Ancestral (black/gray), eutherian-specific (red) and mouse-specific (orange) TSSs and the location of the DNA transposon MER53 (blue) are also shown. **(B)** Diagram showing long and short Mbnl3 isoforms and the location of eutherian-specific aa changes. Zinc finger domains are shown in blue. Sequence alignment is shown in S3 Fig. **(C)** Quantification of the relative usage of the splice junctions highlighted in (A) across a range of mouse tissues. **(D)** Transgenic analysis of a putative Mbnl3 promoter region. A approximately 8-kb region surrounding the eutherian-specific TSS indicated by the black bar in (A) produced lacZ staining in the trophoblast of 4/10 injected embryos at E7.5. **(E)** Phylogenetic tree showing the degree of MBNL3 (red) and

MBNL2 (blue) protein sequence divergence in eutherian (circles) and noneutherian (triangles) species. The tree was produced by the neighbor-joining method using Kimura corrected distances. **(F)** RNAcompete-derived sequence logos for the indicated human (Hs), mouse (Mm), opossum (Op), and chicken (Gg) MBNL proteins. **(G)** Analysis of the Z-score contribution of sequences containing GCUU or GCAGG motifs to the total cumulative Z-score of the top 100 RNAcompete-derived 7-mers for the indicated MBNL proteins. **(H)** Bar plot showing the Z-score of the top RNAcompete-derived 7-mer for each of the indicated MBNL proteins. **(I)** Analysis of the Z-score contribution of sequences containing GCG, GCC, GCA, or GCU motifs to the total cumulative Z-score of the RNAcompete-derived 7-mers containing a single GC dinucleotide among the top 100 7-mers for the indicated MBNL proteins. **(J)** Ratio of the z-score contribution of 7-mer's containing a pair of GC dinucleotides with a 1-bp spacer (GCxGC) versus a 2-bp spacer (GCxxGC). * in panels G–J indicate chimeric proteins. The numerical data underlying this figure can be found in S1 Data. aa, amino acid; MBNL, Muscleblind-like; TSS, transcription start site.

noneutherian (chicken and opossum) MBNL3 proteins using RNAcompete [39–41]. We found that full-length opossum and chicken MBNL3 show a strong GCUU binding preference, similar to the binding preferences that have previously been described for mammalian MBNL1 and MBNL2 and nonvertebrate MBNL proteins using a variety of in vitro and in vivo techniques (Fig 2F and 2G) [38,39,42–44]. In contrast, we found full-length mouse MBNL3 to have overall weaker binding preferences, as measured by the top Z-score values (Fig 2H), and to preferentially bind GCA/U (Fig 2F, 2G and 2I). This motif is similar to that identified by previous CLIP-seq–based studies [38,45,46], confirming the validity of using this in vitro technique to explore MBNL binding preferences. Furthermore, we observed that among 7-mer motifs containing a pair of GC di-nucleotides, opossum and chicken MBNL3 show a strong preference for a pair of uracil's between the GCs, while mouse MBNL3 appears to favor a single adenosine residue in that position (GCUUGC versus GCAGC; Fig 2F, 2G, 2I and 2J). Altogether, these results demonstrate that the binding specificity of MBNL3 has specifically diverged in eutherians.

Given the high prevalence of the short MBNL3 protein isoform in trophoblastic tissues, we also assessed its binding preferences alongside those of artificially created short isoforms for chicken and opossum MBNL3 and for mouse MBNL1 and MBNL2. The short isoform of mouse MBNL3 was found to have similar GCU/A binding preferences to the long isoform suggesting both may bind the same target sites (Fig 2F–2J). Intriguingly, similar binding preferences were also seen for all the artificially created short MBNL isoforms, with the partial exception of opossum MBNL3, which, although to a lesser extent than its long isoform, maintained a GCU preference (Fig 2F–2J). These results are in agreement with previous work suggesting the binding preferences of canonical MBNL proteins are largely driven by ZF2 [37,47,48] and suggest that the changed binding properties of the full-length mouse MBNL3 isoform may be due to a reduction in ZF2 binding dominance likely driven by the eutherian-specific mutations we identified (S3A Fig).

To investigate the molecular basis of the changes in binding preferences seen for the long isoform of mouse MBNL3, we next generated a number of chimeric mouse–opossum proteins. Introducing the 2 opossum zinc finger pairs into mouse MBNL3 resulted in the mouse protein reverting to an ancestral like binding preference (Fig 2G–2J, S3C and S3D Fig). However, in contrast, introducing the mouse zinc finger pairs into opossum MBNL3 only resulted in a modest change in binding preference. Specifically, although the consensus sequence remained strongly GCU-biased (Fig 2I, S3C Fig), motifs containing a GCA sequence contributed to 20.0% of the accumulated Z-score of the top 100 motifs, while this contribution was only 8.1% for the wild-type (WT) opossum MBNL3 protein (S3D Fig). Given that the linker region may also play a role in RNA binding [30,33,34], we made an additional chimeric protein consisting of the opossum protein with the mouse linker. Similar to the opossum MBNL3 with the mouse zinc fingers, this chimeric protein showed a modest increase in GCA preference at the expense of GCU when compared to the WT protein (21.7% versus 8.1%; Fig 2I, S3D Fig). Together, these data suggest that the novel binding preference of eutherian MBNL3 is a result of sequence changes in both the zinc finger and linker regions.

## *Mbnl3* restricts placental growth in mouse development

To explore novel and ancestral physiological roles of *Mbnl3* in the developing placenta, we then made use of a mouse line in which both long and short *Mbnl3* isoforms had been knocked out [49]. In addition to analyzing *Mbnl3* single KOs, we also examined *Mbnl2* KOs [37] and the effect of *Mbnl2/3* double knockout (DKO), since, despite the divergence of *Mbnl3* in eutherians, both proteins share 64% of sequence identity in mouse and show a large degree of expression overlap in differentiated trophoblastic tissues, as assessed by in situ hybridization (Fig 1D) and using a previously published single-cell RNA-seq dataset (S4 Fig [50]). Strikingly, *Mbnl3* KO resulted in a significant increase in placental weight at embryonic day 13.5 (E13.5) and E18.5 (Fig 3A, S5A Fig; $P < 0.001$, Wilcoxon rank-sum tests). Analysis of placentas harvested at earlier stages revealed this weight increase first becomes apparent at E11.5 (S5B Fig). In contrast, no significant changes in placenta weight were seen in DKOs, while placentas from *Mbnl2* KOs were found to be slightly smaller than those from their WT littermates at E13.5 (Fig 3A; $P = 0.036$, Wilcoxon rank-sum test). To further characterize the effects of the KOs on placental development, hematoxylin–eosin staining was performed at E18.5 (Fig 3B), and a number of phenotypic abnormalities of varying severity were observed. In DKOs, these included placental disorganization, with substantial quantities of junctional tissue (spongiotrophoblast and trophoblast glycogen cells) found within the labyrinth zone (Fig 3C) and an increased amount of eosinophilic necrotic tissue within the junctional zone and decidua (Fig 3D). Similar, but much less severe histological abnormalities were observed in *Mbnl2* and *Mbnl3* single KOs (Fig 3B–3D). Furthermore, an increase in the amount of both labyrinth and junctional zone tissue was seen in *Mbnl3* KO placentas, in line with their increased placental weight (S5C–S5E Fig).

To assess how these differences in placenta development may affect placenta function and thus embryo development, we next analyzed embryo size. *Mbnl3* KO alone had no effect on embryo weight, even at late gestational stages (Fig 3E), indicating that the increase in placenta size that occurs on *Mbnl3* KO is accompanied by a reduction in placental efficiency (S5F Fig). In contrast, DKO embryos suffered from intrauterine growth restriction; this was severe at E18.5 ($P < 0.001$, Wilcoxon rank-sum tests), when placental efficiency was seen to be significantly lower than in *Mbnl3* single KOs (S5F Fig; $P = 0.015$, Wilcoxon rank-sum tests), but could be observed from as early as E13.5 (Fig 3E; $P = 0.041$, Wilcoxon rank-sum tests). A modest but significant growth restriction was also seen in *Mbnl2* single KO embryos at E18.5 (Fig 3E; $P = 0.010$, Wilcoxon rank-sum tests). To test if the decreased embryo size seen in the DKOs was a direct result of loss of placental and not embryonic *Mbnl2/3*, we used 2 conditional KO strategies. First, we made use of the X-linked nature of *Mbnl3*, which means that placental expression occurs almost exclusively from the maternal allele in mouse [8]. Therefore, female embryos in which the maternal allele is KO and the paternal allele is WT are effectively KO for *Mbnl3* in the placenta and have mosaic *Mbnl3* expression in embryonic tissues (M3[m/+]; S5G Fig). *Mbnl2* KO embryos with maternal *Mbnl3* KO (M2KOM3[m/+]) showed a similar growth restriction to full DKO embryos (Fig 3F). Additionally, placentas from offspring with maternal Mbnl3 KO alone showed a similar size increase to those from full Mbnl3 KOs (S5H Fig). Secondly, we made use of a paternal *Sox2*-Cre mouse line [51] to specifically knock out *Mbnl2* from the epiblast (M2epi; S5I Fig). This resulted in a similar fetal and placental size reduction to that seen in full *Mbnl2* KOs, suggesting that these phenotypes are at least in part due to a role for this gene in the epiblast and its derivatives and not just in trophoblastic tissues (Fig 3G, S5J Fig). Importantly, unlike with the full *Mbnl2* KOs, *Mbnl3* KO did not accentuate the fetal growth phenotype of epiblast-specific *Mbnl2* KOs (epiDKOs, Fig 3G). Altogether, the data from the conditional KOs show that the strong embryo growth restriction seen in full DKOs is specifically due to a role for *Mbnl2* and *Mbnl3* in the placenta.

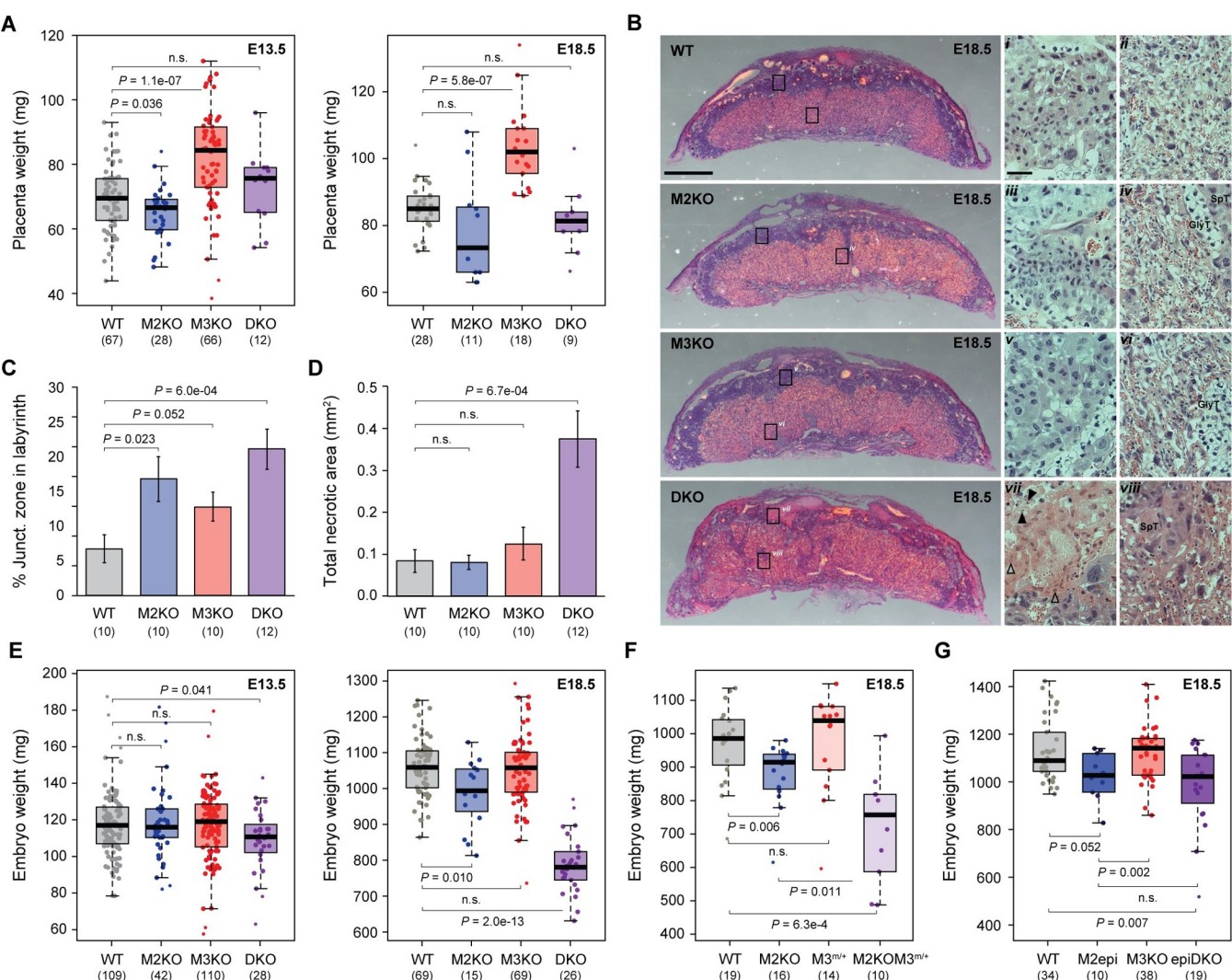

**Fig 3. *Mbnl3* restricts placental growth. (A, E)** Analysis of the effects of M2KO, M3KO, or *Mbnl2:Mbnl3* DKO on placental (**A**) and embryo (**E**) weight at E13.5 and E18.5. (**B**) Hematoxylin–eosin staining of WT and KO placenta sections obtained at E18.5. In the zoomed images, note the GlyT and SpT cells within the labyrinth zone of the KO placentas (iv, vi, viii). Also note the nuclei undergoing karyorrhexis (black arrow head) and karyolysis (empty arrow head) in the DKO junctional zone (vii). Scale bar for full placental sections = 1 mm. Scale bar for zoomed images = 100 μm. (**C, D**) Quantification of the proportion of total junctional zone tissue found within the labyrinth (C) and total area of esophilic necrotic tissue (D) at E18.5. A single medial section was analyzed per placenta. (**F**) Analysis of the effects on embryo weight of placenta specific M3KO with (*M2KOM3^{m/+}*) or without (*M3^{m/+}*) universal M2KO. (**G**) Analysis of the effects on embryo weight of epiblast/fetus specific M2KO with (epiDKO) or without (M2epi) universal M3KO. For (A) and (C–G), the number of placentas/embryos analyzed for each genotype are indicated in brackets, and significance levels are calculated by Wilcoxon rank-sum tests. The numerical data underlying this figure can be found in S1 Data. DKO, double knockout; GlyT, glycogen trophoblast; JZ, junctional zone; LZ, labyrinth zone; M2KO, *Mbnl2* knockout; M3KO, *Mbnl3* knockout; MBNL, Muscleblind-like; SpT, spongiotrophoblast; WT, wild-type.

## *Mbnl2* and *Mbnl3* coregulate alternative splicing and polyadenylation in placenta

To investigate the molecular bases of these phenotypes as well as to gain insight into the mechanism of action of *Mbnl2* and *Mbnl3* in placenta, we next performed RNA-seq of placental samples harvested at E13.5 and E18.5. MBNL proteins have been extensively characterized as regulators of alternative splicing; therefore, we began by using this data to assess the effect of each KO on exon inclusion. At both time points, substantially more and larger splicing changes were seen in DKO than single KO placentas (Fig 4A–4F, S6A–S6C Fig), indicating

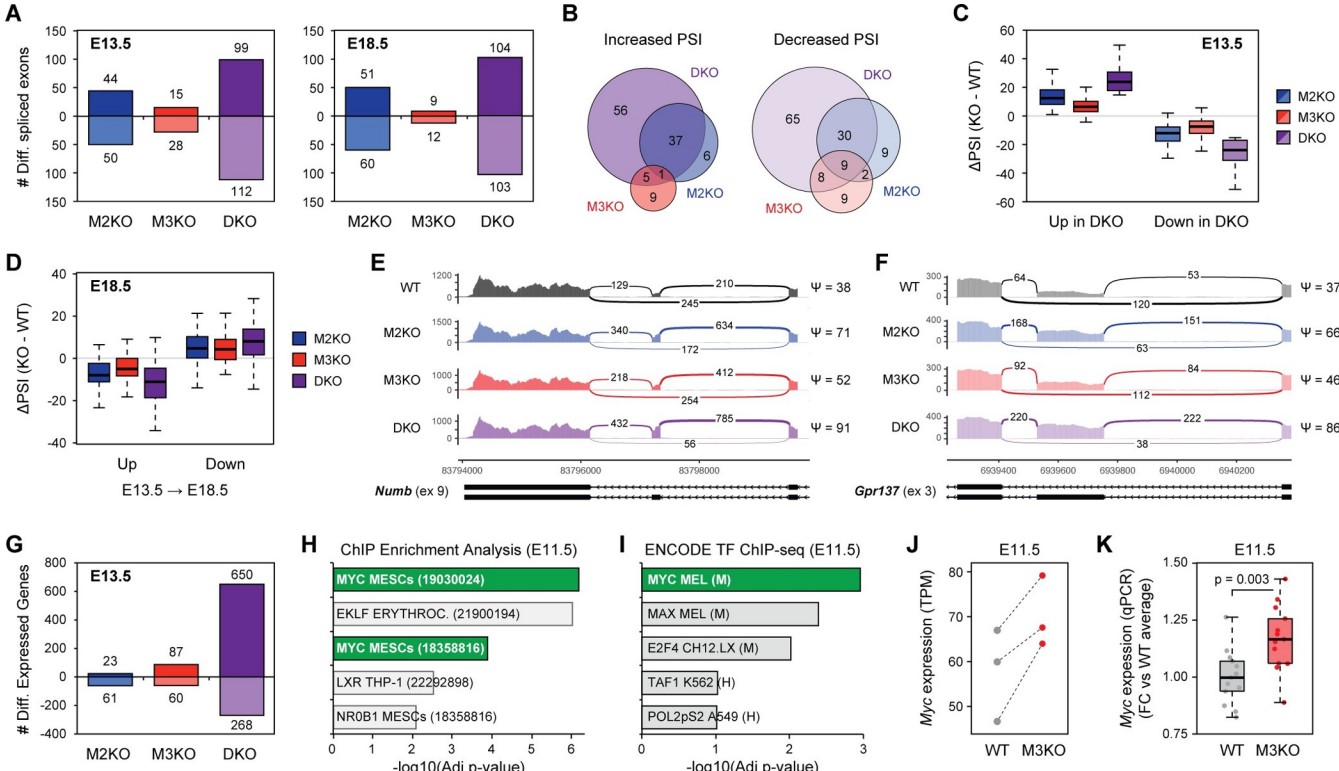

**Fig 4. Effect of *Mbnl2*, *Mbnl3*, and double *Mbnl2/3* KO on the placental transcriptome. (A)** Bar charts showing the number of differentially spliced exons in M2KO, M3KO, and DKO placentas at the indicated embryonic stages. Exons with increased inclusion in KOs are tallied above the x-axis, while exons with reduced inclusion are tallied below. **(B)** Venn diagrams showing the overlap in differential spliced exons between M2KO, M3KO, and DKO placentas at E13.5. **(C)** Boxplots showing the difference in exon inclusion (PSI) in the indicated KOs versus WT placentas for all exons differentially spliced in DKOs versus WTs at E13.5. **(D)** Boxplots showing the change in PSI in the indicated KO versus WT placentas at E18.5 for all exons with differential inclusion between E13.5 and E18.5 WT placenta. **(E, F)** Sashimi plots showing usage of *Numb* exon 9 (E) and *Gpr137* exon 3 (F) in placentas with the indicated genotypes at E13.5. The PSI values for the exons in the different genotypes are shown to the right of the plots. **(G)** Bar charts showing the number of differentially expressed genes in M2KO, M3KO, and DKO placentas at E13.5. Genes with increased expression in KOs are tallied above the x-axis, while genes with reduced expression are tallied below. **(H, I)** Bar charts showing significance levels for enrichment of TF binding sites among genes up-regulated in M3KO placentas at E11.5. The top 5 most significantly enriched TFs are shown for ChEA (H) and ENCODE (I) data sets. **(J, K)** Plots showing *Myc* expression levels in WT and M3KO placentas at E11.5 based on RNA-seq (J) and qPCR (K). In (J), matched placenta pools derived from the same litters are indicated by dashed lines. For (K), values correspond to the fold change between each tested placenta and the average of WT placenta; *N* = 13 placentas for each genotype. Significance levels were calculated by Wilcoxon rank-sum tests. The numerical data underlying this figure can be found in S1 Data. DKO, double knockout; KO, knockout; MBNL, Muscleblind-like; M2KO, *Mbnl2* knockout; M3KO, *Mbnl3* knockout; PSI, percentage spliced in; qPCR, quantitative real-time PCR; RNA-seq, RNA sequencing; TF, transcription factor; WT, wild-type.

that *Mbnl2* and *Mbnl3* regulate an overlapping set of events and partially compensate for one another's loss. In agreement with previous reports [38,46,52], the number and extent of splicing changes in *Mbnl2* KOs were greater than those in *Mbnl3* KOs (Fig 4A–4C, S6A Fig), and no evidence for substantial *Mbnl3*-specific splicing effects was observed. We also found no evidence of *Mbnl3* acting in placenta as a competitive inhibitor of splicing regulation by *Mbnl2* [45,49], as almost all splicing events significantly altered in DKOs were misregulated in the same direction in *Mbnl2* and *Mbnl3* single KOs (Fig 4C). In line with the known function of MBNL proteins in regulating differentiation-associated alternative splicing [53], we found that misregulated exons acquired more immature-like inclusion levels, as shown both by analyzing splicing changes occurring between E13.5 and E18.5 in WT placentas (Fig 4D) and between embryonic stem cells (ESCs) and differentiated tissues (S6B Fig). Functional enrichment analysis for genes containing misregulated exons identified functions related to cell signaling, migration, and adhesion that could relate to the disrupted placental morphology seen in the

DKOs (S7 and S8 Tables). Among the most misregulated events were alternative exons in *Numb* (Fig 4E), *Gpr137* (Fig 4F), and *Numa1* (S6C Fig), whose immature-like inclusion patterns have been shown to inhibit differentiation, decrease epithelial tightness, and associate with increased cell proliferation, respectively [54–57].

In addition to alternative splicing regulation, MBNL proteins have also been reported to modulate alternative polyadenylation site usage [58], and MBNL3, in particular, has been found to preferentially bind the 3′ end of transcripts [46]. Therefore, we next generated 3′ sequencing data from the same set of placenta samples. As with alternative splicing, we found MBNL2 and MBNL3 to redundantly regulate polyadenylation site usage, and their loss to lead to more immature-like usage patterns (S7A–S7D Fig). However, in contrast to alternative splicing, relatively few genes showed significant alterations in polyadenylation site usage at E13.5 (S7A Fig). Among the misregulated targets (S9 Table) was the transcriptional repressor *Mbd2*, which showed an increase in the use of a proximal polyadenylation site from approximately 0% of transcripts in the WTs to approximately 50% in the DKOs (S7E Fig). Use of this site results in the production of a shortened protein isoform that promotes pluripotency in ESCs, whereas the long isoform enhances cell differentiation [59,60]. This result suggests that a similar mechanism likely operates in TE tissue.

## Misregulated gene expression patterns confirm maturation defects in DKO placentas

To further characterize the molecular changes occurring in single and DKO placentas, we next used the RNA-seq data to examine differences in steady-state mRNA levels (hereafter gene expression). Overall, the results from this analysis mirrored those from splicing and polyadenylation site usage, with substantial changes in gene expression being seen in DKOs and only moderate changes in *Mbnl2* and *Mbnl3* single KOs (Fig 4G, S8A–S8C Fig). Maturation defects were also observed at the level of gene expression in E18.5 KO placentas (S8D Fig). Indeed, among the genes up-regulated in DKO placentas at this stage were a number of transcription factors (TFs) associated with TS cell maintenance, including *Esrrb* [61], *Eomes* [62], and *Cdx2* [63] (S10 Table). This was accompanied by a reduction in expression of a number of genes involved in labyrinth formation and maturation, including the transcriptional regulators *Gcm1* [64] and *Cebpa* [65], and the *Gcm1* target *SynB*, which plays a role in labyrinth cell fusion [66,67] (S10 Table). Gene Ontology (GO) analyses of differentially expressed genes revealed multiple functions altered in DKOs, including terms related to cell adhesion, migration, and extracellular matrix organization (S11 Table).

## *Mbnl3* loss leads to up-regulation of Myc and its targets at E11.5

In contrast to alternative splicing and polyadenylation usage, analysis of gene expression at E13.5 revealed a subset of genes that appeared to robustly respond to *Mbnl3* but not *Mbnl2* loss (S8B and S8E Fig). These genes were not significantly misregulated in *Mbnl2* KOs and had similar levels of misexpression in *Mbnl3* KOs and DKOs (S8E Fig, S10 Table). This indicates that eutherian *Mbnl3* has some regulatory activities not possessed by *Mbnl2*, in line with its unique morphological phenotype. Among these genes were a number with a potential role in nutrient uptake, *Slc2a3*, *LipG*, *Atp1b1*, *Steap3*, and *Slc19a2*, which where all up-regulated following *Mbnl3* loss (S10 Table). This would be consistent with the X-linked nature of *Mbnl3* and the expected maternal bias in its regulatory activities resulting from this; however, it could also be a result of the placenta compensating for functional deficiencies resulting from *Mbnl3* loss.

To further investigate the molecular bases behind the increase in placenta growth specifically seen in *Mbnl3* KOs, we performed RNA-seq of *Mbnl3* KO placentas at E11.5. This is the stage when the increase in placental size was first observed (S5B Fig) and a time of proliferative growth for the organ [68,69]. Differential gene expression analysis at this stage revealed 10 genes to be significantly up-regulated and 23 genes to be significantly down-regulated, over half of which came from a single genomic cluster and were members of the carcinoembryonic antigen (CEA) gene family (S10 Table). Extensive examination of the genes showing significantly altered expression did not reveal any obvious candidates that may be responsible for the observed increase in placenta size. Therefore, we next performed a series of functional and regulatory enrichment analyses using looser cutoffs for differential gene expression ($P < 0.1$, log2FC $> 0.1$). These revealed enrichment for GO terms consistent with increased growth among up-regulated genes, including ribosome, translation, and metabolic processes (S12 Table). In addition, they showed that the pro-proliferative TF *Myc* has the most significant level of binding site enrichment in the up-regulated genes according to both ChEA and ENCODE ChIP datasets (Fig 4H and 4I, S12 Table). Similarly, *MycN* was found to be the top candidate in a TF perturbation dataset (S12 Table). In line with these results, expression of *Myc* was up-regulated approximately 1.2 fold at E11.5, both based on RNA-seq data and qPCR assays using independent placenta samples (Fig 4J and 4K). Up-regulation of *Myc* and its associated regulatory network is consistent with an increase in proliferation contributing to the larger placentas seen in *Mbnl3* KO mice.

## *Mbnl3* KO alleviates fetal growth reduction following maternal calorie restriction

The increase in placental growth seen on *Mbnl3* KO is consistent with the theory that the eutherian X chromosome is enriched for placentally expressed genes that favor increased maternal fitness [9–11]. In this context, *Mbnl3* would act by limiting the amount of resources allocated to placenta growth, and thus fetal development. However, while there is an obvious maternal cost in terms of energy expenditure for increasing placenta size in *Mbnl3* KO animals, we did not observe any fetal advantage, such as increased embryo size. This could be either due to a suboptimal placenta function in *Mbnl3* KOs or because embryo size is already optimal under standard laboratory conditions, where mothers have ad libitum access to food; indeed, in normally gown near-term litters from WT mice, there is no correlation between placenta and fetus size [70]. The second possibility led us to hypothesize that the increased placental size seen in *Mbnl3* KOs may provide additional placental reserve capacity to *Mbnl3* KO embryos when resources are limited. To test this, we examined the effects of *Mbnl3* KO on fetal growth following midgestational maternal calorie restriction, which results in a reduction in embryo size of approximately 50% at E18.5 (S9A Fig [71]). Under these conditions, placentas from *Mbnl3* KOs were still found to be larger than those of their WT littermates (Fig 5A and 5B, S9B Fig). However, in contrast to noncaloric restricted conditions and in agreement with our hypothesis, analysis of embryo weights revealed *Mbnl3* KO fetuses from calorie restricted pregnancies to be significantly larger than their WT littermates at E18.5 (Fig 5C and 5D, S9C Fig).

## Discussion

We have identified 2 related RNA-binding proteins, MBNL2 and MBNL3, with differing evolutionary histories that play both overlapping and distinct roles in placenta development. Despite its accelerated sequence evolution, *Mbnl3* appears to have retained sufficient ancestral regulatory activity to act partially redundantly with *Mbnl2* to control placental morphology

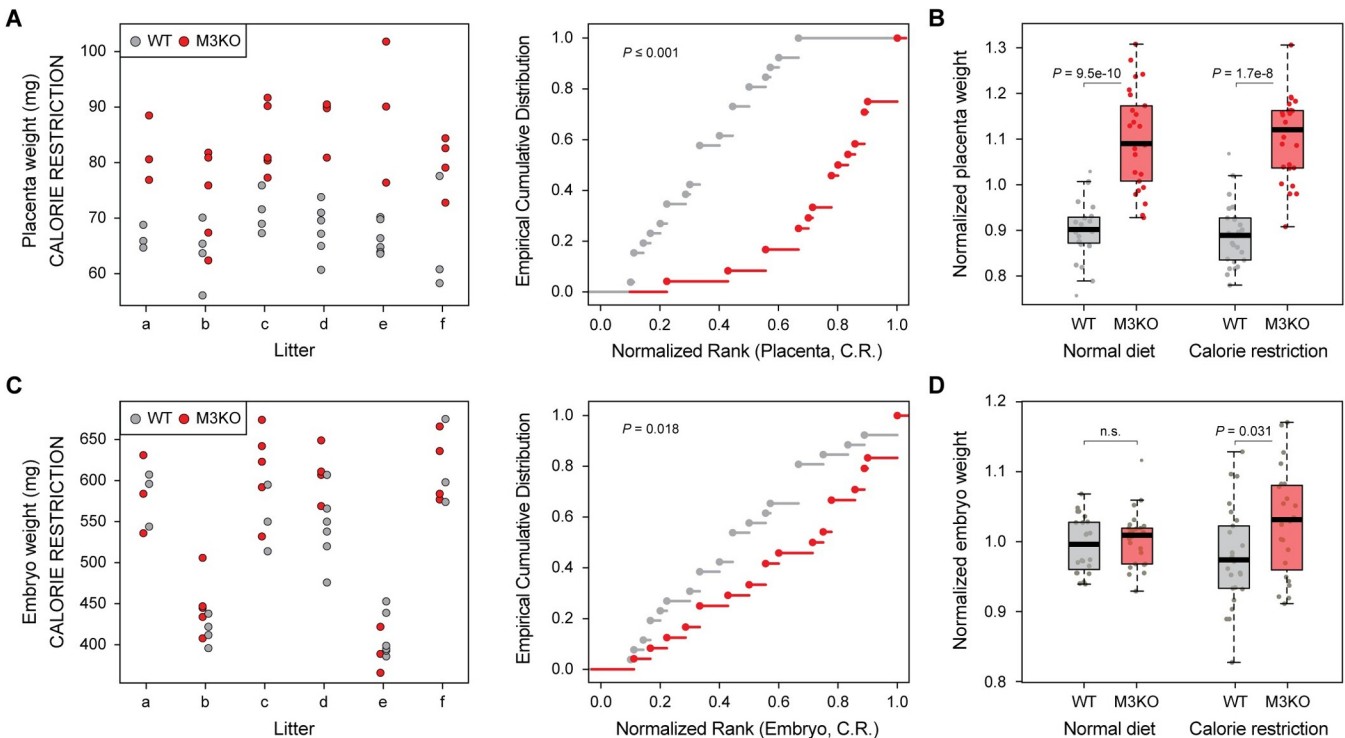

**Fig 5. M3KO reduces the effect of midgestational maternal calorie restriction on fetal growth. (A, C)** Dot plots showing the weights of individual WT (gray) and M3KO (red, M3KO) placentas (A) and embryos (C) harvested from 6 calorie restricted mothers and the corresponding empirical cumulative distribution plot for each genotype. Significance levels were calculated by a permutation test with 1,000 iterations swapping the genotype labels within each litter. **(B, D)** Analysis of the effects of maternal calorie restriction on the weight of WT and M3KO placentas (B) and embryos (D) at E18.5. For each embryo/placenta, the weight was normalized with respect to the average of WT and KO weights within each litter. Significance levels are calculated by Wilcoxon rank-sum tests. The numerical data underlying this figure can be found in S1 Data. KO, knockout; M3KO, *Mbnl3* knockout; WT, wild-type.

and physiology and thus facilitate proper fetal growth. In particular, *Mbnl2* and *Mbnl3* appear to be required for proper placenta transcriptomic maturation and organization of the labyrinth and junctional zones and to prevent excessive cell death/necrosis at the maternal fetal interface. The need for MBNL proteins for transcriptomic maturation in placenta is in line with that observed in other tissues and likely occurs via both co- and posttranscriptional regulation of mRNA metabolism, particularly alternative splicing [37,53,58,59,72–75].

On the other hand, *Mbnl3* has acquired a new biological function limiting placental growth. The increase in placental size in mouse *Mbnl3* KOs is first seen at around E11.5. This is a time when placental growth is largely driven by cell proliferation and not an increase in cell size, which occurs from E12.5 onward [68,69]. Furthermore, the pro-proliferative TF *Myc* is up-regulated in *Mbnl3* KOs at E11.5, and genes up-regulated in these KOs are enriched for *Myc* binding within their promoters. Altogether, this suggests that *Mbnl3* limits placental growth at least in part by reducing cell proliferation in association with lower levels of *Myc* expression. The putative effects of *Mbnl3* on *Myc* levels are, however, unlikely to be direct, as previous CLIP-seq experiments in other cell types with robust *Myc* expression have found no evidence that MBNL3 binds directly to *Myc* transcripts [46,58]. Therefore, it remains unclear how *Mbnl3* exerts its effects on placental growth; however, it is tempting to speculate that it relates to the evolution of its novel RNA-binding specificity and/or of a eutherian-specific short isoform with a single pair of zinc fingers (as discussed in S1 Text). Previous studies have found the 2 *Mbnl3* isoforms to be in distinct cytoplasmic compartments, with the long isoform largely

localized to ribosomes and the short isoform largely soluble, suggesting that they play distinct roles in RNA metabolism [46]. Given that both isoforms are strongly expressed in mouse placenta and have similar binding preferences that are distinct from those of canonical MBNL proteins, they could modulate different aspects of the metabolism of the same RNA transcripts in a coordinated manner to inhibit growth. However, the short isoform appears to be more abundant in both cow and humans, hinting that it may harbor the majority of this activity. Irrespectively, the important regulatory interactions likely occur in the cytoplasm as eutherian *Mbnl3* has lost an alternative exon driving nuclear localization and has a more cytoplasmic localization, a higher ratio of 3′ UTR to intronic binding, and is a weaker splicing regulator than other MBNL proteins (this paper and [38,46]).

Although both *Mbnl2* and *Mbnl3* appear to have been recruited to the placenta early in eutherian evolution, this recruitment has been accompanied by loss of expression from other tissues and neofunctionalization only for *Mbnl3*, but not for *Mbnl2*. This is likely linked to *Mbnl3*'s genomic location on the X chromosome. Due to paternal-specific inactivation in placenta, the mouse X chromosome is expected to accumulate alleles that favor the maternal side of the parental conflict [9–11]. The increase in placenta size seen in *Mbnl3* KOs is in keeping with this hypothesis, in terms of the associated maternal cost of the higher energy requirements. However, the increase in placental size seen in *Mbnl3* KOs was accompanied by a reduction in placental efficiency. This appears to be a common phenomenon since increased placental size is associated with reduced placental efficiency both in WT animals and when placental overgrowth results from gene KO, as is the case for the X linked genes *Esx1*, *Plac1*, and *Nrk* and the paternally silenced autosomal genes *Phlda2* and *Cdkn1c* [12–14,70,76,77]. While the reduced efficiency seen in the gene KOs may, in some circumstances, be due to placental defects, the results from WT animals suggests that, under standard laboratory conditions, placental efficiency is largely driven by fetal demand. Indeed, although KO of the paternally silenced genes *Igf2r*, *Grb10*, and *Cdkn1c* and of the X-linked gene *Nrk* results in increased fetal growth in addition to placental growth, these genes are substantially expressed in both the fetus and placenta [13,77–79], making the causal connection between placental and embryo size unclear.

Unlike in those KO models, *Mbnl3* is largely not expressed in the embryo. However, we show here that, despite the reduced placental efficiency, the increased placenta size seen in *Mbnl3* KOs results in better maintenance of fetal growth following maternal calorie restriction. This demonstrates that *Mbnl3* KO can confer a paternal advantage in circumstances that could conceivably occur frequently in an animal's natural environment. Altogether, this suggests an evolutionary model where changes to the *Mbnl3* loci leading to reduced placental growth have been favored due to its genomic location and have either driven and/or been facilitated by its tissue-restricted expression, consolidating *Mbnl3* as a novel player in the parental genetic conflict in eutherians.

## Methods

### Identification of splicing factors enriched in trophoblastic tissues

We used a list of previously defined RNA-binding proteins and other genes known to be involved in splicing and/or alternative splicing regulation [25,26]. We restricted the analysis to 197 splicing factors with one-to-one orthologs between mouse and human based on Ensembl BioMart. One-to-one orthologs versus mouse were also retrieved for cow and opossum, when available. All studied splicing factors for the 4 species are listed in S13 Table.

To investigate the enrichment of these splicing factors in trophoblastic tissues in mouse, we focused on 3 comparisons: (i) placenta versus nonplacental tissues; (ii) trophoblast versus ICM

at the blastocyst stage; and (iii) TS cells versus embryonic and XEN stem cells. Gene expression quantifications, using the transcript per million (TPM) metric, for public RNA-seq data for different placenta and nonplacenta tissues as well as embryonic blastomeres and ESCs were obtained from *VastDB* [80], assembly mm10. Short Read Archive (SRA) identifiers and associated information are listed in S1 Table. In addition, for comparison (iii), we generated de novo RNA-seq of different lines of cultured TS cells and XEN stem cells. Culture details and sequencing and mapping information are provided below and in S1 Table. TPM values for TS and XEN samples were obtained using *vast-tools* v2.5.1 [80].

For comparison (i), for each gene, we first calculated the average TPM expression for each tissue type (groupings are provided in the column "Group" of S1 Table). Next, we calculated the log2 fold change of expression between placental tissues and the average of nonplacental tissues, which are plotted in Fig 1A and shown in S2 Table. Only genes with TPM > 10 in placental tissues were considered. For the other 2 comparisons (Fig 1B and 1C), we followed a similar logic and averaged the expression of each gene in (ii) TE and ICM blastomeres at E3.5 and (iii) TS cells and the average of ESCs and XEN stem cells (ESC and XEN).

To compare placental and nonplacental tissues in human, cow, and opossum, we used a similar approach to the one described for mouse, relying on public RNA-seq data. For human (hg38) and cow (bosTau6), we retrieved TPM values from VastDB [80]. For opossum and nonmammalian vertebrates, we collected RNA-seq data from the SRA (S1 Table) and calculated TPMs using *vast-tools* v2.5.1 for each species.

## Single-cell transcriptomic analysis of Mbnl2 and Mbnl3 expression in the placenta

Single-nuclei RNA-seq data were obtained from [50]. The Seurat object containing processed and integrated data from all trophoblast nuclei sequenced at different time points (E9.5, E10.5, E12.5, and E14.5) was downloaded from figshare (https://figshare.com/projects/Single_nuclei_RNA-seq_of_mouse_placental_labyrinth_development/92354). Data from individual time points were extracted from the joint Seurat object using the subset function from the Seurat package [81]. For each time point, we generated UMAP plots showing the distribution of the cells of each sample across clusters. Additionally, we generated *Mbnl2* and *Mbnl3* co-expression plots using the FeaturePlot function with the parameters blend = TRUE, order = T, blend.threshold = 0, and split violin plots using a custom extension of ggplot2 GeomViolin function [82] to generate split violins of the density distribution of *Mbnl2* and *Mbnl3* expression in each individual cluster.

## Analysis of Mbnl3 transcription start sites and alternative splicing

To identify potential TSSs and alternative splicing events leading to different amino-terminal isoforms of MBNL3, we looked for competing splicing donors (5′ splice sites) that are spliced to the splicing acceptor (3′ splice site) of the second coding exon, which contains the in-frame ATG giving rise to the short protein isoform and that is present in all known *Mbnl3* transcripts. For this purpose, we took advantage of *vast-tools align*, which generates an output file for each RNA-seq sample (sample.eej2) containing the read counts for all exon–exon junctions annotated in the *vast-tools* junction-based library [80]. The reference splicing acceptor for *Mbnl3* for each species were the following: 11 from ENSMUSG00000036109 (mm10), 16 from ENSG00000076770 (hg38), and 0 from ENSBTAG00000014088 (bosTau6). The corresponding donors are numbered starting from 0 (the most upstream) for each species, and the coordinate on the X chromosome and its location in the genome are provided in S2A–S2C Fig and S6 Table. Each donor was grouped and colored based on the evolutionary origin of the associated

TSS region and whether or not it would lead to a short MBNL3 protein isoform: black, long isoform from the ancestral TSS; gray, short isoform from the ancestral TSS by skipping of the first coding exon; red, short isoform from the eutherian-specific TSS; and orange, short isoform from the rodent/primate-specific TSSs. To provide the percent of reads coming from each group of donors (Fig 2C, S2E and S2F Fig), we calculated the percent of reads for each of those groups out of the total number of exon–exon junction reads to the reference acceptor as provided in the sample.eej2 files, and, then, we grouped similar samples (e.g., neural tissues, muscle, etc.) as indicated in the column "GROUP" of S6 Table.

## Western blotting

Western blot analysis was conducted according to standard laboratory procedures. Lysates were prepared from staged placental samples by disruption with a sterile pestle in Laemmli buffer (0.05M Tris-HCl at pH 6.8, 1% SDS, 10% glycerol, 0.1% b-mercaptoethanol) on ice. Samples were then boiled at 95˚C for 10 minutes, sonicated using a bioruptor (10 cycles, 30 seconds on, 30 seconds off), and spun at 20,000 g for 90 seconds. Proteins were separated by 4–15% SDS-PAGE gel (Bis-Tris Criterion TGX pre-cast gels, Bio-Rad, USA) and transferred to PVDF membranes (Thermo Scientific, USA). Membranes were incubated overnight with MBNL3 antisera [46] at 1:500 or anti Lamin B1 (1:1,000, Cell Signaling Technology, USA) in 5% nonfat dairy milk in PBS plus Igepal (0.05%, Merck, USA).

## Mouse transgenic reported assays

Three fragments of the putative enhancer/promoter region surrounding the eutherian-specific TSS of *Mbnl3* were amplified from mouse genomic DNA using the primers shown in S14 Table. The 3 fragments were then cloned together into a single modified pBluescript vector [83] containing a lacZ reporter gene. Prior to microinjection, this construct was linearized and plasmid sequences removed. The construct was then microinjected at 3 to 6 ng/μl into the pronucleus of fertilized mouse oocytes at E0.5. Embryos were collected from E6.5 to E7.5, fixed, and stained for β-galactosidase activity.

## Molecular evolution analyses for Mbnl2 and Mbnl3

Protein sequence alignments and tree production were made using the CLC main work bench (ver. 5.5.1, QIAGEN, USA) software package. MBNL isoforms used for alignment only contained the coding exons common to all *Mbnl2* and *Mbnl3* orthologs (coding exons 5 [54 bp] and 9 [approximately 74 bp] were excluded from all aligned sequences). Deeply conserved residues (red box, S3A Fig) were defined as those conserved across MBNL2 and noneutherian MBNL3 proteins, whereas all other residues were considered nondeeply conserved (green boxes, S3A Fig). The protein tree was constructed by the neighbor-joining method with Kimura protein corrections. To investigate loss of coding exon 5 (54 bp), *Mbnl2* and *Mbnl3* orthologs were first examined on the Ensembl (version 102) and UCSC (http://genome.ucsc. edu) genome browsers. For *Mbnl* genes where exon 5 was not annotated, exon loss was confirmed by manual inspection of the intronic region.

## MBNL constructs used for RNAcompete

For all *Mbnl* constructs tested, the exon usage pattern followed that of the predominant MBNL3 isoform in mouse placenta: All MBNL proteins tested included alternative coding exon 8 (95/98 bp) and lacked alternative coding exons 5 (54 bp), 7 (36 bp), and 9 (approximately 74 bp). Mouse *Mbnl* isoforms were cloned from placenta/limb cDNA using the primers

detailed in S14 Table. DNA sequences corresponding to opossum and chicken MBNL isoforms alongside the chimeric MBNLs were synthesized in vitro by Bio Basic and IDT, respectively. Corresponding short isoforms were generated either from genomic DNA or cloned long isoforms using the primers detailed in S14 Table.

### RNAcompete and associated bioinformatic analyses

The RNA pool generation, RNAcompete pull-down assays, and microarray hybridizations were performed as previously described [39–41]. For the pull down assays, GST-tagged RNA-binding proteins (20 pmoles) and RNA pool (1.5 nmoles) were incubated in 1 mL of Binding Buffer (20 mM Hepes pH 7.8, 80 mM KCl, 20 mM NaCl, 10% glycerol, 2 mM DTT, 0.1 mg/mL BSA) containing 20 mL glutathione sepharose 4B (GE Healthcare, USA) beads (prewashed 3 times in Binding Buffer) for 30 minutes at 4˚C and subsequently washed 4 times for 2 minutes with Binding Buffer at 4˚C.

One-sided Z-scores were calculated for all 7-mer motifs as described previously [39]. Z-scores approximate the affinity of the RNA-binding protein for that specific 7-mer. To obtain a measure of the relative contribution of a specific motif (e.g., GCUU or GCAGC) to the overall affinity, we summed the Z-scores of all 7-mers containing the motif among the top 100 7-mers and calculated the fraction of that sum with respect to that of all 100 7-mers (Fig 2G). Similarly, to assess the relative contribution of the nucleotide after a GC, for each 7-mer containing a GC among the top 100 7-mers, we summed the Z-scores for 7-mers containing GCA, GCC, GCG, or GCU and calculated the fraction corresponding to each of them (Fig 2I). Finally, for those 7-mers containing GCxGC or GCxxGC among the top 100 7-mers, we calculated the relative contribution of those with GCxGC or GCxxGC motifs as the log2 ratio between the sums of their Z-scores (Fig 2J).

### Animal husbandry and embryo and placenta weight analysis

All protocols were carried out in accordance to the European Community Council Directive 2010/63/EU and approved by the local Ethics Committee for Animal Experiments (Comitè Ètic d'Experimentació Animal-Parc de Recerca Biomèdica de Barcelona, CEEA-PRBB, CEEA 9086 and MDS 0035P2). $Mbnl3^{tm2.1Sws/+}$ [49], $Mbnl2^{tm1.1Sw/+}$ [37], $Mbnl2^{tm1Sws/+}$ [37], and $Edil3^{Tg(Sox2-cre)1Amc}$ [51] mice alongside the required intercrosses were maintained on mixed backgrounds. Embryo genotyping was conducted on tail biopsies as described [84] using published primers. Embryo sex determination was based on the presence or absence of the Y-linked gene *Sry* as determined by PCR using primers detailed in S14 Table.

In all analyses, WT littermates were used as controls. Furthermore, female *Mbnl3* heterozygous embryos carrying a WT maternal allele were considered to be WT for *Mbnl3* as it is almost exclusively expressed from the maternal allele in placenta (S5F Fig), and *Mbnl2* heterozygotes were considered WT for *Mbnl2* as neither placenta nor embryo size was significantly altered in *Mbnl2* heterozygotes versus full WT embryos regardless of the KO status of *Mbnl3*. For the analysis of placenta weights from full KO embryos at E18.5, data were combined from crosses generating just constitutive KO embryos and crosses generating embryos with placenta specific *Mbnl3* KO in addition to constitutive KO embryos; therefore, some of the data from these crosses was used in both Fig 3A and S4H Fig. For the analysis of placenta specific *Mbnl3* KOs (a genotype only obtainable in female embryos) all results from male siblings of all genotypes were excluded.

For placental weight analysis at E10.5 and E11.5, only the embryonic portion of the placenta was analyzed, which was separated from the maternal decidua by blunt dissection. For placentas analyzed from E13.5 onward, the whole placenta was weighed.

## Maternal calorie restriction analysis

For maternal calorie restriction experiments, pregnant dams were fed 50% of their normal daily chow intake from E11.5 until embryo harvesting [71]. Only litters with at least 3 WTs and 3 KOs were used for the analyses. Raw values for placenta and embryo weight for each genotype within each litter are provided in Fig 5A and 5C (calorie restricted diet) and S9B and S9C Fig (normal diet). To normalize for the large variability observed among litters, we performed 2 complementary approaches. First, we plotted the values as empirical cumulative distribution function (ECDFs) to visualize the effects. To test the hypothesis that KO placentas/embryos are heavier than their littermates, we ranked each placenta/embryo within each litter and obtained a total sum of ranks per genotype within each litter. We then randomly swapped the genotype labels within each litter 1,000 times and computed in how many iterations the sum of ranks was the same or smaller than in the test case. Second, we normalized each placenta/embryo measure by dividing its weight by the average of the mean WT and mean KO weights within each litter (Fig 5B and 5D). Statistical significance of the difference between the normalized values was assessed using Wilcoxon rank-sum tests.

## Whole-mount in situ hybridization and hematoxylin–eosin staining

Whole-mount in situ hybridization was carried out following standard procedures [85]. Probes were synthesized from cDNAs cloned into the pGEM-T Easy (Promega, USA) plasmid. The cloning primers used for probe generation are shown in S14 Table. Hematoxylin–eosin staining was conducted according to standard laboratory practice on 6-μm sagittal midsections from placentas harvested at E18.5, fixed overnight in 4% PFA and embedded in paraffin wax. Sections from 10 or more placentas were examined per genotype.

Images of whole embryos and placenta sections were captured using an Olympus SZX16 stereo microscope (Olympus, Germany) and DP73 digital camera. Zoomed images were captured using a Leica DM6000 B upright microscope (Leica, Germany) and DFC420 digital camera with a 20× objective. For quantification of hematoxylin–eosin stained sections, labyrinth, junctional (spongiotrophoblast and glycogen trophoblast), and neurotic tissue was manually defined based on morphology. Analysis was conducted using ImageJ (Version 1.50g, NIH, Maryland, USA). A single medial section was quantified per placenta. Total labyrinth tissue area = (total area of labyrinth zone)–(total area of junctional tissue in the labyrinth). Total junctional tissue area = (area of junctional zone) + (area of junctional tissue in the labyrinth).

## RNA sample collection for sequencing and RT-qPCR

TS[JR] [27], TS[FxL4] [86], XEN[JR] [87], and XEN[DIZ6] [86] cell lines were cultured as described [27,87]. For E11.5 placental samples, the whole embryonic portion of the placentas was separated from the maternal decidua by blunt dissection. For E13.5 and E18.5 placental samples, broad medial sections of placenta were cut and tissue layers separated using forceps. For E13.5 samples, the embryonic portion of the placenta was collected. For E18.5 samples, only the labyrinth zone was collected to avoid signal from the necrotic cells found in the junctional zone of DKO placentas (Fig 3B and 3D) overwhelming the analysis. For sequencing of E11.5 placentas, 3 samples, each containing 3 placentas, were collected per genotype, and samples were paired so that WTs and mutants came from the same litter(s). For sequencing of E13.5 and E18.5 placentas, 2 samples, each containing tissue from a minimum of 5 placentas, were collected for each genotype. In all cases, RNA was extracted using the RNeasy MiniKit (QIAGEN).

## Quantitative real-time PCR

For qPCR, cDNA was prepared from total RNA using Superscript III reverse transcriptase (Invitrogen, Lithuania) and random nonamer primers (Invitrogen). qPCR was performed using the NZYSpeedy qPCR Green Master Mix (NZYTech, Portugal). Primers used are detailed in S14 Table. For analysis of *Mbnl2* and *Mbnl3* expression, the housekeeping genes *Ywhaz* and *Hmbs* were used for normalization. For analysis of *Myc* expression, the housekeeping genes *Ywhaz*, *Srp14*, *Gapdh*, and *Eef2K* were used for normalization, and expression was calculated relative to average WT expression.

## Next-generation sequencing

For gene expression and alternative splicing analyses, standard polyA-selected Illumina libraries were generated using standard protocols by the CRG Genomics Unit. An average of approximately 100 million (XEN and TS cells), approximately 47 million (E13.5 and E18.5) 125-nt paired end, or approximately 47 million 50-nt single end (E11.5) reads were generated using a HiSeq2500 machine. For alternative polyadenylation analysis, the same E13.5 and E18.5 RNA samples were processed by the CRG Genomics Unit with the QuantSeq 3′ mRNA-Seq Library Prep Kit Reverse (REV) from Lexogen (Austria) following the manufacturer's instructions. An average of approximately 11 million 50-nt single end reads were generated using a HiSeq2500 machine. General and mapping statistics for all samples are provided in S1 Table.

## Alternative splicing analysis

To obtain exon inclusion levels, we processed RNA-seq samples with *vast-tools* v2.5.1 [80] following default options. In brief, each sample was mapped using *vast-tools align* and the mm10 VASTDB library (vastdb.mm2.23.06.20.tar.gz). Next, tables with inclusion levels using the percentage spliced in (PSI) metric for all exons were generated for E13.5 and E18.5 samples separately using *vast-tools combine* with the default options. Events with sufficient read coverage across all samples within a given time point were then extracted using *vast-tools tidy* with the following options:—min_N 8—min_SD 0—noVLOW (i.e., only exons with LOW or higher coverage scores were allowed). From these exons, differentially spliced exons between each KO and the WT for each time point were defined as those with an average absolute change in PSI (ΔPSI) between KO and WT ≥15 and at least ΔPSI ≥5 between each of the 2 replicates in the same direction. GO enrichment analyses for genes with differentially spliced exons for each KO versus WT comparison at each time point were performed using DAVID v6.8 [88] for GOTERM_BP_DIRECT, GOTERM_MF_DIRECT, GOTERM_CC_DIRECT, and KEGG_PATHWAY. As a background set for each time point, we used genes with at least 1 exon event annotated in *vast-tools* with sufficient read coverage across all samples as defined above (9,211 genes for E13.5 and 8,263 for E18.5). Sashimi plots for selected events were generated using *ggsashimi* [89] with BAM files produced by mapping RNA-seq reads to mm10 genome using STAR [90].

Exons that change during placental development were defined as those with a significant change between WT samples at E13.5 and E18.5 based on the same cutoffs (|average ΔPSI| ≥ 15 and ΔPSI between each replicate ≥5 in the same direction). ESC-differential exons (i.e., exons that are differentially spliced between ESCs and differentiated cell and tissue types) were defined as previously reported [53]. For this purpose, we obtained PSI values from *VastDB* for mouse (mm10 release 21/12/21) and obtained the average ΔPSI between ESCs and differentiated cell and tissue types using the script Get_Tissue_Specific_AS.pl [91] (https://github.com/vastdb-pastdb/pastdb) with the config file provided in S15 Table and the option—test_tis ESC.

## Polyadenylation site usage analysis

We aligned the Lexogen Quantseq Reverse sequences to the mm11 assembly (Ensembl 104 gene annotation) using STAR with parameters—outFilterMultimapNmax 1—outFilter-MatchNminOverLread 0.2—outFilterScoreMinOverLread 0.2—sjdbGTFfile. We then clustered 5-end of alignments with inverted strand (with polyA signal filtering to avoid most internal priming events) to construct the polyA database from our data. We then used this database to compute polyA site counts (clustering of alignments around detected polyA sites with 50 down/up stream nt tolerance) and used DEXSeq [92] to identify regulated polyA sites in genes. We report regulated genes with polyA site pairs that undergo fold changes in opposite directions and for which changes are significant (false discovery rate [FDR] < 0.1).

For further analyses, we selected a single polyA pair per gene and stage (S9 Table). For this purpose, among the pairs (if more than 1) that were significantly misregulated in the largest number of KO versus WT comparisons (from 0 to 3), we performed a ranked selection: (i) we took the pair with the largest absolute fold change if the total number of reads across the 4 conditions for the pair were at least 15% of the pair with the largest number of reads; (ii) else, we took the pair with the largest number of reads of all (not only significant) pairs; (iii) else, if the pair had more than 30% of the number of reads of the pair with the largest number of reads, we took the one with the highest fold change; and (iv) else, we discarded the gene.

## Gene expression analysis

RNA-seq samples were used to estimate transcript abundances using *Salmon* quasi-mapping approach with default parameters plus the—seqBias and the—gcBias options to perform sequence-specific and GC bias correction, respectively, and the—validateMappings option to ensure that mappings are validated using an alignment-based verification [93]. The reference fasta file used for the quantification was the set of cDNA sequences for Ensembl and ab initio genes predicted in the mm10 genome assembly in Ensembl [94] (Version 88). Gene expression estimates were obtained from transcript abundance estimates using the txtimport package [95] with the option countsFromAbundance = lengthScaledTPM, to obtain estimated gene counts scaled using the average transcript length over samples and the library size.

Differential gene expression analysis was performed using these gene expression estimates with the *DESeq2* package [96] with default settings. In WT versus KO differential expression analyses, genes with an FDR-adjusted *P* value (padj) smaller than 0.05 (E13.5 and E18.5 samples) or 0.1 (E11.5 samples) were considered differentially expressed. For display purposes, shrunken log2 fold changes were used. To investigate the effect of the different KOs on placenta maturation, log2 fold changes for each KO at E18.5 were plotted for genes that were differentially expressed between E13.5 and E18.5 in WT samples using strict cutoffs (padj < 0.01 and |FC| > 5) (S8D Fig).

GO analyses for differentially expressed genes for each KO at E13.5 and E18.5 were performed using DAVID v6.8 [88] for GOTERM_BP_DIRECT, GOTERM_MF_DIRECT, GOTERM_CC_DIRECT, and KEGG_PATHWAY. Genes quantified by *DESeq2* were provided as background set (17,488 for E13.5 and 17,221 for E18.5). For E11.5, to increase the list of potentially differentially expressed genes, we relaxed the cutoffs to select genes with *P* < 0.1 and |log2FC)| > 0.1 (409 and 331 up- and down-regulated genes, respectively). We conducted GO enrichment analyses for these subsets of genes using DAVID as described above with all genes for which a valid *P* value (non-NA) is proved by DESeq2 as background set (16,972 genes). In addition, we used the gene set enrichment analyses for TF binding (ChEA_2016 and ENCODE_TF_ChIP-seq_2015) and the "TF Perturbations Followed by Expression" test from the web tool *EnrichR* [97].

## Supporting information

**S1 Text. Supporting information discussion.**
(PDF)

**S1 Fig.** *Mbnl3* **expression is highly enriched in placenta in eutherians. (A)** Whole-mount in situ hybridisation analysis of *Esrp1* in mouse embryos at the indicated stages. **(B)** Box plot showing expression levels of *Esrp1* in placenta and nonplacental tissues in mouse. **(C, D)** Scatter plots showing placental expression level and placenta versus other tissues enrichment for 197 splicing regulators in human (C) and in cow (D). **(E)** Expression of *Mbnl3* orthologs from different vertebrate species across differentiated adult tissues. Placenta samples are highlighted in red. RNA-seq samples are listed in S1 Table. The numerical data underlying this figure can be found in S1 Data. RNA-seq, RNA sequencing.
(TIF)

**S2 Fig. Novel TSSs in Mbnl3, short isoform usage, and placental *cis*-regulatory elements. (A–C)** Genome organization for mouse (A), human (B), and cow (C) of the annotated donor splice sites upstream of the second coding exon of *Mbnl3*, which encodes the internal ATG used to translate the short isoform. The coordinate on the X chromosome is indicated for each competing donor. The color code corresponds to that of Fig 2A, and the donor IDs correspond to that of S6 Table. **(D)** Western blot showing short (approximately 27 kDa) and long (approximately 38 kDa) MBNL3 isoforms in placenta at different developmental time points. The blot was conducted using an antisera raised against a 15-amino acid carboxyl-terminal MBNL3 peptide [21]. **(E, F)** Percent of exon–exon junction reads from each group of competing donors (as shown in A–C and schematized in the legend) across different samples from human (E) and cow (F). Individual donor counts for each sample are listed in S6 Table. **(G)** UCSC Genome Browser view of the mouse *Mbnl3* genomic locus showing the injected promoter region (black box—yellow highlight). Top 5 tracts generated by Starks and colleagues [22] using E9.5 placenta ChIP-seq data. Track 5 ("Placenta CREs") shows ChromHMM-defined chromatin states, including repressed regions (light blue), poised enhancers (dark green), active enhancers (red), and active promoters (light purple). Note, the injected promoter region was cloned as 3 fragments (see Methods) that were then ligated together to make the final tested construct; a portion of the locus was not included in this construct due to low levels of conservation with other placental mammals as indicated by the gap in the black box. The numerical data underlying this figure can be found in S1 Data. MBNL, Muscleblind-like; TSS, transcription start site.
(TIF)

**S3 Fig. Molecular evolution of eutherian MBNL3 proteins. (A)** Sequence alignment for different MBNL3 (red) and MBNL2 (blue) proteins from various vertebrate species. Eutherian changes in deeply conserved (conserved between MBNL2 and noneutherian MBNL3 proteins) and nondeeply conserved residues are highlighted using red and green boxes, respectively. ZF and the liker region are indicated. **(B)** Presence (black) or absence (white) of exon 5 in different *Mbnl* genes across vertebrates. In opossum, exon 5 can be identified, but it contains an in-frame stop codon (gray). **(C)** RNAcompete-derived sequence logos for the indicated chimeric proteins, with either mouse (Mm) or opossum (Op) ZF or linker regions in the other species' backbone. **(D)** Analysis of the Z-score contribution of sequences containing GCA motifs to the total cumulative Z-score of the top 100 RNAcompete-derived 7-mers for the indicated WT and chimeric MBNL proteins. The numerical data underlying this figure can be found in S1 Data. MBNL, Muscleblind-like; WT, wild-type; ZF, zinc finger.
(TIF)

**S4 Fig. Single-cell transcriptomic analysis of *Mbnl2* and *Mbnl3* expression during placenta development.** For each developmental stage, top, UMAP plots showing the distribution of cells across clusters (left) and the expression of *Mbnl2*, *Mbnl3*, and their co-expression in individual cells (right) obtained from [23]. The color code of the cells in the co-expression plot is defined according to the expression level of *Mbnl2* and *Mbnl3* as displayed in the color scale. Bottom, split violin plots showing the expression distribution of *Mbnl2* and *Mbnl3* for each cell cluster. The numerical data underlying this figure can be found in S1 Data. JZP 1/2, Junctional zone precursors subtype 1 and 2; LaTP, labyrinth trophoblast progenitor; LaTP 2, labyrinth trophoblast progenitor subtype 2; SpT, spongiotrophoblast; SynTI, syncytiotrophoblast layer 1; SynTII, syncytiotrophoblast layer 2; S–TGC, sinusoidal trophoblast giant cells.
(TIF)

**S5 Fig. Mbnl3 restricts placental growth. (A)** Representative images of whole placentas at E18.5 for the indicated genotypes. **(B)** Effect of M3KO on placental weight at E10.5 and E11.5. **(C–E)** Quantification of the total area of labyrinth tissue (C), total area of junctional zone tissue (D) and ratio between labyrinth and junctional zone tissues for the indicated genotypes. **(F)** Effect of M2KO, M3KO or *Mbnl2:Mbnl3* DKO on placenta efficiency (embryo weight/placenta weight) at E13.5 (top) and E18.5 (bottom). **(G, I)** *Mbnl3* (F) and *Mbnl2* (G) expression quantified by qPCR in the indicated genotypes. Each bar corresponds to a single placenta/limb sample. **(H)** Effects on placenta weight of placenta specific *Mbnl3* knockout with ($M2KOM3^{m/+}$) or without ($M3^{m/+}$) universal M2KO. Note, some of these data points are also plotted in Fig 3A. **(J)** Effects on placenta weight of epiblast/fetus specific M2KO with (epiDKO) or without (M2epi) universal *Mbnl3* knockout. For (B–F, H, and J), the number of placentas/embryos analyzed for each genotype are indicated in brackets and significance levels are calculated by Wilcoxon rank-sum tests. The numerical data underlying this figure can be found in S1 Data. DKO, double knockout; M2KO, *Mbnl2* knockout; M3KO, *Mbnl3* knockout; qPCR, quantitative PCR.
(TIF)

**S6 Fig. Effect of *Mbnl2*, *Mbnl3* and double *Mbnl2/3* knockout on alternative splicing and *Numa1* exon 15 inclusion. (A)** Venn diagrams showing the overlap in differential spliced exons between M2KO, M3KO, and DKO placentas at E18.5. **(B)** Boxplots showing the change in PSI in the indicated KO versus WT placentas at E18.5 for all exons with differential inclusion between ESCs and differentiated cell and tissues. **(C)** Sashimi plots showing usage of *Numa1* exon 15 in placentas with the indicated genotypes at E13.5. The PSI values for the exons in the different genotypes are shown to the right of the plots. The numerical data underlying this figure can be found in S1 Data. DKO, double knockout; M2KO, *Mbnl2* knockout; M3KO, *Mbnl3* knockout; PSI, percentage spliced in; WT, wild-type.
(TIF)

**S7 Fig. Effect of *Mbnl2*, *Mbnl3*, and double *Mbnl2/3* knockout on alternative polyadenylation. (A, B)** Bar charts showing the number of genes with differentially used alternative polyadenylation (polyA) sites in M2KO, M3KO, and DKO placentas at E13.5 (A) and E18.5 (B). Genes whose proximal polyA site increased usage in KOs are tallied above the x-axis while genes with reduced proximal polyA usage are tallied below. **(C)** Boxplots showing the difference in proximal polyA site usage (log2 FC) in the indicated KOs versus WT placentas for all genes with differentially used polyA sites in DKOs versus WTs at E18.5. Up in DKO, $N = 124$; down in DKO, $N = 318$. **(D)** Boxplots showing the change in proximal polyA site usage in the indicated KO versus WT placentas at E18.5 for all genes polyA site pairs that are differentially

used between E13.5 and E18.5 WT placenta. Up in E18.5, *N* = 31; down in E18.5, *N* = 77. **(E)** Sashimi plots showing usage of the 2 competing polyA sites in *Mbd2* in placentas with the indicated genotypes at E13.5 based on 3′-seq data. The PU values for the proximal polyA site in the different genotypes are shown to the right of the plots. Coordinates correspond to mm10. The numerical data underlying this figure can be found in S1 Data. DKO, double knockout; KO, knockout; M2KO, *Mbnl2* knockout; M3KO, *Mbnl3* knockout; PU, percent of usage; WT, wild-type.
(TIF)

**S8 Fig. Effect of *Mbnl2*, *Mbnl3*, and double *Mbnl2/3* knockout on gene expression. (A)** Bar charts showing the number of differentially expressed genes in M2KO, M3KO, and DKO placentas at E18.5. Genes with increased expression in KOs are tallied above the x-axis, while genes with reduced expression are tallied below. **(B)** Venn diagrams showing the overlap in differentially expressed genes between M2KO, M3KO, and DKO placentas at E13.5. **(C)** Box-plots showing the difference in gene expression (log2 FC) in the indicated KOs versus WT placentas for all genes differentially expressed in DKOs versus WTs at E13.5. **(D)** Boxplots showing the change in expression in the indicated KO versus WT placentas at E18.5 for all genes differentially expressed between E13.5 and E18.5 WT placenta. **(E)** Dot plots showing the change in expression in the indicated KO versus WT placentas at E13.5 for genes that are differentially expressed only in M3KOs and DKOs. The numerical data underlying this figure can be found in S1 Data. DKO, double knockout; FC, fold change; M2KO, *Mbnl2* knockout; M3KO, *Mbnl3* knockout; WT, wild-type.
(TIF)

**S9 Fig. M3KO reduces the effect of midgestational maternal calorie restriction on fetal growth. (A)** Weight distributions of WT placentas (top) and embryos (bottom) under normal or calorie restricted diets. **(B, C)** Dot plots showing the weights of individual WT (gray) and M3KO (red, M3KO) placentas (B) and embryos (C) harvested from 5 mothers fed with N.D. and the corresponding empirical cumulative distribution plot for each genotype. Significance levels were calculated by a permutation test with 1,000 iterations swapping the genotype labels within each litter. The numerical data underlying this figure can be found in S1 Data. M3KO, *Mbnl3* knockout; N.D., normal diet; WT, wild-type.
(TIF)

**S1 Table. RNA-seq data used in this study.** For each species, SRA identifiers, read number, read length and source are provided for each public RNA-seq file used in this study. Groups used to identify placenta-enriched factors are also provided. For RNA-seq samples generated in this study, mapping statistics are also provided. RNA-seq, RNA sequencing; SRA, Short Read Archive.
(XLSX)

**S2 Table. Mouse splicing factors enriched in trophoblastic tissues.** log2 FCs and expression (log10TPM) in the trophoblastic/placenta tissue is provided for mouse splicing factors for each comparison: (i) placenta versus tissues; (ii) TE versus ICM; and (iii) TS cells versus ESCs and XEN stem cells. Only genes with a minimal expression of TPM $\geq$ 10 in the target tissue are included. ESC, embryonic stem cell; FC, fold change; ICM, inner cell mass; TE, trophecto-derm; TPM, transcript per million; TS, trophoblast stem; XEN, extraembryonic endoderm.
(XLSX)

**S3 Table. Human splicing factors enriched in placenta.** log2 FC between placenta and non-placental tissues and expression (log10TPM) in the placenta tissue is provided for human

splicing factors with an expression in placenta of TPM ≥ 10. FC, fold change; TPM, transcript per million.
(XLSX)

**S4 Table. Cow splicing factors enriched in placenta.** log2 FC between placenta and nonplacental tissues and expression (log10TPM) in the placenta tissue is provided for cow splicing factors with an expression in placenta of TPM ≥ 10. FC, fold change; TPM, transcript per million.
(XLSX)

**S5 Table. Opossum splicing factors enriched in placenta.** log2 FC between placenta and nonplacental tissues and expression (log10TPM) in the placenta tissue is provided for opossum splicing factors with an expression in placenta of TPM ≥ 10. FC, fold change; TPM, transcript per million.
(XLSX)

**S6 Table. Usage of TSSs in eutherian *Mbnl3*.** For each sample and species, number of reads and percent over the total mapping to each competing splice junction are provided. Competing junctions consist of alternative splice donors (normally resulting from alternative promoters) joint to the same splice acceptor in the third ancestral exon (see scheme in Fig 2A). Each donor annotated in *vast-tools* is numbered, starting from 0, and the coordinate on the X chromosome is provided in the header (e.g., 3:132413583; donor 3, coordinate chrX:132413583). The coordinates and donor numbers are depicted in S2A–S2C Fig for each species. The color code for each column corresponds to the scheme of Fig 2A: black, canonical long isoform using the first ATG-encoding exon; gray, ancestral TSS skipping the first ATG-encoding exon; red, eutherian-specific TSS; orange, rodent or primate specific TSS. The column "Group" shows the sample groupings using to generate the quantifications displayed in Fig 2C and S2E and S2F Fig. TSS, transcription start site.
(XLSX)

**S7 Table. Differentially spliced exons.** Exons found to be differentially spliced in at least 1 KO versus WT comparison at the E13.5 or E18.5 stage. Gene ID, gene symbol, VastID, genomic coordinate (mm10), and exon length are provided alongside the inclusion level (PSI) for each WT and KO replicate and the change in developmental time (PSI_E18.5—PSI_E13.5). Key: The exon is up-regulated (U), down-regulated (D), or not differentially spliced (N) in the KO versus WT comparison for the DKO, M2KO and M3KO (e.g., DDN, down-regulated in DKO and M2KO but not M3KO). DKO, double knockout; KO, knockout; M2KO, *Mbnl2* knockout; M3KO, *Mbnl3* knockout; PSI, percentage spliced in; WT, wild-type.
(XLSX)

**S8 Table. Enriched GO terms for genes with differentially spliced exons.** Significantly enriched terms for each KO versus WT comparison at E13.5 or E18.5 as provided by DAVID. Only GOTERM DIRECT (biological process, molecular function, and cellular component) and KEGG pathways were tested. GO, Gene Ontology; KO, knockout; WT, wild-type.
(XLSX)

**S9 Table. Pairs of alternative polyA site analyzes in this study.** For E13.5 or E18.5, a pair of alternative polyA sites per gene was selected (see Methods for details). Genomic features, raw read counts, and log2 FCs for each KO versus WT (KO/WT) comparison and E18.5 versus E13.5 using the proximal polyA site as reference are provided. Enhanced/Repressed indicates a given polyA pair is significantly differentially regulated in the KO and the direction refers to the proximal polyA, e.g., an enhanced pair means the proximal site is up-regulated in the KO. KO_Up/KO_Down indicates that the proximal site is up/down-regulated in the KO, but not

reaching statistical significance. FC, fold change; KO, knockout; WT, wild-type.
(XLSX)

**S10 Table. Differentially expressed genes.** For each stage and KO comparison, the log2 FC and FDR-adjusted *P* value (padj) are provided for each expressed gene. Mbnl3-specific genes at E13.5 are defined as those differentially expressed in the M3KO and DKO comparisons but not in the M2KO one. Genes with a potential role in nutrient uptake are highlighted in green. DKO, double knockout; FC, fold change; FDR, false discovery rate; KO, knockout; M2KO, *Mbnl2* knockout; M3KO, *Mbnl3* knockout.
(XLSX)

**S11 Table. Enriched GO terms for differentially expressed genes.** Significantly enriched terms for up- and down-regulated genes for each KO versus WT comparison at E13.5 or E18.5 as provided by DAVID. Only GOTERM DIRECT (biological process, molecular function, and cellular component) and KEGG pathways were tested. GO, Gene Ontology; KO, knockout; WT, wild-type.
(XLSX)

**S12 Table. Enriched GO terms and EnrichR results for differentially expressed genes at E11.5.** Significantly enriched terms for up- and down-regulated genes in M3KO versus WT at E11.5 as provided by DAVID are shown. Only GOTERM DIRECT (biological process, molecular function, and cellular component) and KEGG pathways were tested. The results for the regulatory enrichment using EnrichR are also provided for up-regulated genes. GO, Gene Ontology; M3KO, *Mbnl3* knockout; WT, wild-type.
(XLSX)

**S13 Table. Splicing regulators.** List of 197 splicing regulators used in this study and Ensembl IDs for one-to-one orthologs in human, mouse, cow and opossum.
(XLSX)

**S14 Table. Primers used in this study.** Restriction sites added to primers and used in cloning are highlighted in red (enhancer fragments #1, SacII and NotI, #2, SalI and SalI, #3 NotI and NotI. Mbnls for RNA compete, AscI and SbfI). MBNL, Muscleblind-like.
(XLSX)

**S15 Table. Config file for calling ESC-differential exons.** Config file used to run Get_Tissue_Specific_AS.pl to calculate ΔPSI values (ESC—other cell/tissue types). ESC, embryonic stem cell.
(XLSX)

**S1 Data. Excel spreadsheet containing the underlying numerical data for Figs 1A, 1B, 1C, 1E, 1F, 2C, 2G, 2H, 2I, 2J, 3A, 3C, 3D, 3E, 3F, 3G, 4C, 4D, 4H, 4I, 4J, 4K, 5A, 5B, 5C and 5D and S1B, S1C, S1D, S1E, S2E, S2F, S3D, S4, S5B, S5C, S5D, S5E, S5F, S5G, S5H, S5I, S5J, S6B, S7C, S7D, S8C, S8D, S8E, S9A, S9B, and S9C Figs.**
(XLSX)

**S1 Raw images. Original western blot images for S2D Fig.**
(PDF)

## Acknowledgments

We thank Ben Blencowe for initial support on this project and Luis P. Iñiguez for help on generating plots. Animal silhouettes were downloaded from http://phylopic.org. We acknowledge

the support of the CERCA Programme/Generalitat de Catalunya and of the Spanish Ministry of Economy, Industry and Competitiveness (MEIC) to the EMBL partnership.

## Author Contributions

**Conceptualization:** Thomas Spruce, Demian Burguera, Moises Mallo, Barbara Pernaute, Manuel Irimia.

**Data curation:** Thomas Spruce.

**Formal analysis:** Mireya Plass, André Gohr, Gregor Rot, Hong Zheng, Manuel Irimia.

**Funding acquisition:** Manuel Irimia.

**Investigation:** Thomas Spruce, Mireya Plass, André Gohr, Debashish Ray, Ana Nóvoa, Demian Burguera, Manuel Irimia.

**Methodology:** Thomas Spruce, Debashish Ray, María Martínez de Lagrán, Gregor Rot, Ana Nóvoa, Jon Permanyer, Marta Miret, Moises Mallo, Mara Dierssen, Barbara Pernaute, Manuel Irimia.

**Resources:** Debashish Ray, Maurice S. Swanson, Quaid Morris, Mara Dierssen, Timothy R. Hughes, Barbara Pernaute, Manuel Irimia.

**Software:** Gregor Rot.

**Supervision:** Quaid Morris, Mara Dierssen, Timothy R. Hughes, Barbara Pernaute, Manuel Irimia.

**Validation:** Thomas Spruce, Jon Permanyer, Moises Mallo.

**Visualization:** Thomas Spruce, André Gohr, Moises Mallo, Manuel Irimia.

**Writing – original draft:** Thomas Spruce, Manuel Irimia.

**Writing – review & editing:** Debashish Ray, Maurice S. Swanson, Barbara Pernaute, Manuel Irimia.

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
