## [Editor Report · Decision Letter 0]

27 Oct 2021

Dear Dr Irimia, 

Thank you for submitting your manuscript entitled "The X-linked splicing regulator MBNL3 has been co-opted to restrict placental growth in eutherians" for consideration as a Research Article by PLOS Biology.

Your manuscript has now been evaluated by the PLOS Biology editorial staff, as well as by an academic editor with relevant expertise, and I am writing to let you know that we would like to send your submission out for external peer review.

IMPORTANT: In its current form, we are not wholly convinced that the advance will be sufficient. However, we propose that we review it as a Discovery Report (https://journals.plos.org/plosbiology/s/what-we-publish#loc-discovery-report), with the hope that a future paper might supply the mechanism by which MBNL3 regulates MYC. However, if the reviewers do not show sufficient enthusiasm, we could explore the possibility of a transfer to PLOS Genetics. No format changes are needed at this time, but please change the article type to "Discovery Report" when you upload your additional metadata (see next paragraph).

Once your full submission is complete, your paper will undergo a series of checks in preparation for peer review. Once your manuscript has passed the checks it will be sent out for review. 

If your manuscript has been previously reviewed at another journal, PLOS Biology is willing to work with those reviews in order to avoid re-starting the process. Submission of the previous reviews is entirely optional and our ability to use them effectively will depend on the willingness of the previous journal to confirm the content of the reports and share the reviewer identities. Please note that we reserve the right to invite additional reviewers if we consider that additional/independent reviewers are needed, although we aim to avoid this as far as possible. In our experience, working with previous reviews does save time. 

If you would like to send your previous reviewer reports to us, please specify this in the cover letter, mentioning the name of the previous journal and the manuscript ID the study was given, and include a point-by-point response to reviewers that details how you have or plan to address the reviewers' concerns. Please contact me at the email that can be found below my signature if you have questions. 

Please re-submit your manuscript within two working days, i.e. by Oct 29 2021 11:59PM.

Kind regards,

Roli Roberts

Roland Roberts

Senior Editor

PLOS Biology

rroberts@plos.org

---

## [Decision Letter · Decision Letter 1]

7 Jan 2022

Dear Dr Irimia,

Thank you for submitting your manuscript "The X-linked splicing regulator MBNL3 has been co-opted to restrict placental growth in eutherians" for consideration as a Discovery Report at PLOS Biology. Your manuscript has been evaluated by the PLOS Biology editors, an Academic Editor with relevant expertise, and by two independent reviewers. We had recruited a third reviewer, but they were unable to provide their assessment in a timely fashion.

You'll see that both of the reviewers were broadly positive about your study, but each raises a number of concerns that should be addressed before further consideration. Each of them worries that the mechanistic link to Myc is rather circumstantial, but we had recognised this limitation prior to review, and consider that as beyond the scope of a Discovery Report. Reviewer #2 only has one minor request, but while many of reviewer #1's requests are textual, some their points will require additional analyses and/or data.

In light of the reviews (below), we will not be able to accept the current version of the manuscript, but we would welcome re-submission of a much-revised version that takes into account the reviewers' comments. We cannot make any decision about publication until we have seen the revised manuscript and your response to the reviewers' comments. Your revised manuscript is also likely to be sent for further evaluation by the reviewers.

IMPORTANT: As well as addressing the reviewers' concerns, we note that you currently have 5 main Figures. The maximum allowable for a Discovery Report is 4, so please could you either combine some of the Figures, and/or move some material to the Supplementary Figures?

We expect to receive your revised manuscript within 3 months. 

**IMPORTANT - SUBMITTING YOUR REVISION**

*Re-submission Checklist*

*Published Peer Review*

*PLOS Data Policy*

*Blot and Gel Data Policy*

Sincerely,

Roli Roberts

Roland Roberts

Senior Editor

PLOS Biology

rroberts@plos.org

REVIEWERS' COMMENTS:

Reviewer #1:

[identifies himself as Andrew Ward]

This is an excellent study of Mnbl3 gene function that uncovers its role in trophoblast and placental development, evolutionary changes in the regulation of the gene and in RNA-binding preference of MBNL3 protein, as well as potential molecular mechanisms of action. This adds up to a well-rounded and convincing story. The transcriptomic data reveals changes in Myc regulated gene expression that could make a major contribution to the Mbnl3 KO phenotype, though there are currently 'disconnects' in that there is no evidence Myc is directly regulated by MBNL3, and Myc regulated increases in cell proliferation are inferred but not proven. The paper contains a great deal of novel data and makes an important contribution in the fields of placental development, function and evolution. There are one or two pieces of data that appear to be missing, these and my other detailed comments and suggestions are presented below.

Introduction

Nice and concise, covering the relevant background.

Results

P4 and Fig 1, language a bit imprecise, e.g. tissue stages not given for placenta and non-placenta samples; it would be helpful to indicate that only splicing factors were plotted and how many (197 according to the Methods); Is it true to say "Expression enrichment in the trophoblast for the top candidates was confirmed by in situ hybridization analysis" or rather something along the lines that "Consistent with the enrichment analysis, expression each of the top candidates was readily detected in the trophoblast by in situ hybridization analysis"?

P4-5, How has the trophoblast stem cell compartment been defined?

P5, "both full-length and truncated MBNL3 protein isoforms were present in mouse placenta by Western blot analysis (Fig. S2d)". It would be useful to know whether the MBNL3 antisera used to probe Westerns is known to recognise the C-terminus of the protein. This could be stated here or in the Methods or legend to Fig. S2d.

P5 and Fig 2d, The transient mouse transgenic experiment is covered in a somewhat cursory fashion. Three putative promoter/enhancer fragments are mentioned in the Methods but the only data is a single LacZ stained embryo (representative of 4/10 examined, Fig 2d). I wonder if there is more to be learned from the full dataset? At the least the fragments could be better defined, perhaps alongside some bioinformatic analysis of conserved elements or hypersensitive sites.

P5-7, I have no particular issues with the phylogenetic sequence analyses or the MBNL3 binding studies, where the conclusions seem entirely commensurate with the data. The group have extensive prior experience with the RNAcompete method but it should be noted as a caveat that the conclusions are dependent on this one method.

P7, "a modest [change] in…"?

P8, Both Mbnl3 and Mbnl2 are said to be expressed in trophoblastic tissues. A reference to Fig 1d would be useful here but also references to any information on their expression as the placenta develops. To what extent does their expression overlap in trophoblastic tissues through development?

P9, placental phenotypes are said to be due to Mbnl2/3 genes acting "partly redundantly in this tissue to ensure proper placental development". There is a lot of nice data presented on this but whether the genes act partly redundantly is not entirely clear (the trancriptomic data lends some support later on and perhaps this conclusion should be reserved for the Discussion). There seems to be an additive effect on placental size, with Mbnl2 KO causing size reduction and Mbnl3 an expansion. This adds up to a size similar to WT placentae in the double KO, but with the DKO placenta exhibiting a much more disorganised morphology than either single KO. This suggests the genes are acting largely independently of one another and perhaps while the placenta can compensate functionally for loss of either one, loss of both is more catastrophic.

P9, in addition to the plots of embryo and placenta size (Fig. 3 and Fig S4) it may be useful to plot placental efficiency (fetal mass as a function of placental mass). This would highlight that although enlarged the Mnbl3 KO placenta is inefficient (see Discussion point below).

P9, Inclusion of the conditional KO strategies is very nice but in both cases the placental weight data is missing. In the case of paternal transmission of Mnbl3 to produce Mnbl3+/p offspring, are the placentae large as in the homozygotes? In contrast, for the epiblast-specific KOs, was placental size roughly normal? Incidentally, here (including in Fig S4) the nomenclature could be tightened up as convention dictates that the alleles are written maternal/paternal and where it is relevant to indicate which parental allele is mutant this can be expressed as m/+ or +/p.

P9, "At both time points substantially more and larger splicing changes were seen in DKO than single KO placentas" and "number and extent of splicing changes"; I can see the evidence substantiates greater numbers of changes in DKO placentae but I am not seeing in addition a 'larger' or greater 'extent' of changes.

P10-11, there is a general problem in interpreting the transcriptomic data that is common to all KO experiments of this kind in that it is difficult to resolve changes that are a cause or consequence of the resulting phenotype. Logically, any causal changes should occur before the primary phenotype emerges. Here, transciptomic data was collected at multiple stages with the earliest data from e11.5 placentae, at around the same time changes in Mbnl3 KO placenta size was first detected. This builds a picture that Myc-driven expansion of the labyrinth could underlie the primary phenotype in Mbnl3 KO placentas. A Myc:Mbnl3 double KO would help to pin this down conclusively, but may be difficult to perform because of the embryo lethal phenotype of Myc homozygous KOs. However, the case could be strengthened by addressing when and where MYC expression is increased in the trophoblast and through showing that there are associated increases in cell proliferation (increased proliferation is alluded to on P12 but not shown).

P11, The following statement may need further thought: "Consistent with the X-linked nature

of Mbnl3 and the expected resulting maternal bias as well as the increase in placenta size,

amongst these genes were a number with a potential role in nutrient uptake, Slc2a3, LipG,

Atp1b1, Steap3 and Slc19a2, which where all up-regulated following Mbnl3 loss". Once the placenta has become dysmorphic it seems likely that expression of genes involved in nutrient uptake regulation will compensate for these deficiencies, as has been shown in other KO models. Without evidence that MBNL3 directly regulates the indicated genes the evolutionary argument is weakened somewhat in this instance.

P12, the lack of an increase in embryo growth in association with the expanded placental size, and labyrinthine volume in particular, is a bit surprising but not entirely without precedent (see for example global versus placenta-specific KO of Igf2). The inclusion of the maternal gestational caloric restriction experiment is therefore very interesting. The effect on embryo size is subtle and because of the way the data have been normalised it is only possible to see the impact of caloric restriction on embryo size in the box-plot of Fig S8a . It is not clear why there is no equivalent plot for placentae. More generally, this experiment throws up a number of questions that seem worthy of further study (e.g. what happens to nutrient uptake gene expression, do the embryo weight gains persist into post-natal life?). but I am not suggesting they necessarily should be a part of this paper.

Discussion

Arguments are generally concise and balanced. A lot of important discussion points are made in the Results section. Here there is some additional speculation of the molecular mechanism of MNBL3 action and on the evolution of Mnbl3 given its X-linked status. The letter point could perhaps be better made by alluding to placental efficiency, rather than size, and to think about the placenta contributing to producing a fetus that is of an optimal size at term. The authors might also draw on other examples of any other X-linked genes or maternally imprinted genes (e.g. Cdkn1c, Grb10, Igf2r) known to limit growth of the embryo and/or placenta.

Methods

P17, It is not clear what "(red box)" and "(green boxes)" refer to without a figure reference.

Figure legends

S5, Include 'Numa1' in the title line.

Reviewer #2:

Spruce and colleagues present an exploration of the X-linked gene MBNL3 in the control of eutherian placental growth. The placenta is a transient organ that mediates maternal-offspring interactions and is therefore the site of conflict between maternally- and paternally-expressed genes, reflecting differences in the most effective reproductive strategies of males and females. As a consequence, the X chromosome - inherited matrilineally - is enriched for genes that play key roles in placental development and function. 

Most studies examining the mechanisms of placental development and evolution have focused on the level of gene expression, but co- or post-transcriptional processing (splicing, polyadenylation) is likely to play an important role. The authors set out to test this hypothesis. They begin in an unbiased way by comparing the expression of known regulators of splicing between trophoblastic and embryonic tissues in mice, and identify Mbnl3 as being the most enriched regulator in the placenta. The authors then go on to examine the evolution of this gene and determine that it has acquired a novel function in the eutherian placenta. The study has implications for understanding: 1) how the placenta has co-opted ancestral genes to enable it to perform its unique functions in supporting embryonic development; 2) the molecular and genetic basis for that process; and 3) how the genome responds to conflict over resource allocation. The study has broad interest to the fields of embryonic development, evo devo, and the regulation of co-transcriptional processes.

The authors present an excellent manuscript. It is well-written and would carefully guide any reader who is not an expert in this area through the results and conclusions. The study itself is well-designed and the authors use an innovative combination of approaches to reach their conclusion. Throughout the paper, the authors develop hypotheses based on their observations and then test these empirically using a diverse set of tools including transgenic reporter assays and knockout mouse lines. They validate many of their key findings by independent methods - for example, RNA-seq experiments were validated by qRT-PCR on independent samples; observations from sequence data were validated by in situ hybridization and transgenic reporter assays; and results from systemic mouse KO models are validated by using a placenta-specific KO. Overall, the approaches used are thorough, rigorous and robust. 

The weakest part of the paper is the association between Mbnl3 and Myc. The authors clearly realize this and do not over-interpret their findings. Although a mechanistic link between these two genes would strengthen the paper further, it is already a tour de force in evolutionary development, and I recognize that this is beyond the scope of the current manuscript. 

I have only one very minor suggestion for the authors to consider. In Figure 2, it is not immediately clear which sets of data refer to the chimeric proteins. I suggest indicating in the figure which are the chimeric proteins, perhaps with a line underneath their labels to indicate this. As an alternative, providing a key to the naming of the proteins in the figure legend would help.

---

## [Editor Report · Decision Letter 2]

10 Mar 2022

Dear Dr Irimia,

Thank you for submitting your revised Discovery Report entitled "The X-linked splicing regulator MBNL3 has been co-opted to restrict placental growth in eutherians" for publication in PLOS Biology. The Academic Editor and I have assessed your responses and revisions.

Based on this assessment, we will probably accept this manuscript for publication, provided you satisfactorily address the following data and other policy-related requests:

a) We note your arguments for retaining 5 Figures; this is more than is typically allowed for the Discovery Report format, but we're happy to make an exception in this instance.

b) Please could you supply a blurb, as stipulated in the submission form?

c) Please address my Data Policy requests below; specifically, we need you to supply the numerical values underlying Figs 1ABCEF, 2CGHIJ, 3ACDEFG, 4ACDGHIJK, 5ABCD, S1BCDE, S2EF, S3D, S4, S5BCDEFGHIJ, S6B, S7ABC, S8ACDE, S9ABC. Please also cite the location of the data clearly in each relevant main and supplementary Fig legend, e.g. “Data underlying this Figure can be found in S1 Data”.

We expect to receive your revised manuscript within two weeks. 

*Published Peer Review History*

*Press*

Sincerely,

Roli Roberts

Senior Editor,

rroberts@plos.org,

PLOS Biology

DATA POLICY:

Regardless of the method selected, please ensure that you provide the individual numerical values that underlie the summary data displayed in the following figure panels as they are essential for readers to assess your analysis and to reproduce it: Figs 1ABCEF, 2CGHIJ, 3ACDEFG, 4ACDGHIJK, 5ABCD, S1BCDE, S2EF, S3D, S4, S5BCDEFGHIJ, S6B, S7ABC, S8ACDE, S9ABC. NOTE: the numerical data provided should include all replicates AND the way in which the plotted mean and errors were derived (it should not present only the mean/average values).

We require the original, uncropped and minimally adjusted images supporting all blot and gel results reported in an article's figures or Supporting Information files. We will require these files before a manuscript can be accepted so please prepare and upload them now. Please carefully read our guidelines for how to prepare and upload this data: https://journals.plos.org/plosbiology/s/figures#loc-blot-and-gel-reporting-requirements 

DATA NOT SHOWN?

---

## [Editor Report · Decision Letter 3]

29 Mar 2022

Dear Manu,

On behalf of my colleagues and the Academic Editor, Rocio Rivera, I'm pleased to say that we can in principle accept your Discovery Report "The X-linked splicing regulator MBNL3 has been co-opted to restrict placental growth in eutherians" for publication in PLOS Biology, provided you address any remaining formatting and reporting issues. These will be detailed in an email that will follow this letter and that you will usually receive within 2-3 business days, during which time no action is required from you. Please note that we will not be able to formally accept your manuscript and schedule it for publication until you have completed any requested changes.

Sincerely, 

Roli

Roland G Roberts, PhD 

Senior Editor 

PLOS Biology

rroberts@plos.org